# Automatic design of basin-specific drought indexes for highly regulated water systems

Marta Zaniolo[1], Matteo Giuliani[1], Andrea Francesco Castelletti[1,2], and Manuel Pulido-Velazquez[3]

[1]Department of Electronics, Information and Bioengineering, Politecnico di Milano, Piazza L. da Vinci, 32, I-20133 Milano, Italy
[2]Institute of Environmental Engineering, ETH, Wolfgang-Pauli-Str. 15, CH-8093 Zurich, Switzerland
[3]Research Institute of Water and Environmental Engineering (IIAMA), Universitat Politècnica de València, Camí de Vera s/n, 46022, Valencia, Spain

**Correspondence:** Andrea Castelletti (andrea.castelletti@polimi.it)

**Abstract.** Socio-economic costs of drought are progressively increasing worldwide due to undergoing alterations of hydro-meteorological regimes induced by climate change. Although drought management is largely studied in the literature, traditional drought indexes often fail in detecting critical events in highly regulated systems, where natural water availability is conditioned by the operation of water infrastructures such as dams, diversions, and pumping wells. Here, ad-hoc index formulations are usually adopted based on empirical combinations of several, supposed-to-be significant, hydro-meteorological variables. These customized formulations, however, while effective in the design basin, can hardly be generalized and transferred to different contexts. In this study, we contribute FRIDA (FRamework for Index-based Drought Analysis), a novel framework for the automatic design of basin-customized drought indexes. In contrast to ad-hoc, empirical approaches, FRIDA is fully-automated, generalizable, and portable across different basins. FRIDA builds an index representing a surrogate of the drought conditions of the basin, computed by combining all the relevant available information about the water circulating in the system identified by means of a feature extraction algorithm. We used the Wrapper for Quasi Equally Informative Subset Selection (W-QEISS), which features a multi-objective evolutionary algorithm to find Pareto-efficient subsets of variables by maximizing the wrapper accuracy, minimizing the number of selected variables, and optimizing relevance and redundancy of the subset. The preferred variable subset is selected among the efficient solutions and used to formulate the final index according to alternative model structures. We apply FRIDA to the case study of the Jucar river basin (Spain), a drought-prone, highly regulated Mediterranean water resource system, where an advanced drought management plan relying on the formulation of an ad-hoc State Index is used for triggering drought management measures. The State Index was constructed empirically with a trial-and-error process begun in the '80s and finalized in 2007, guided by the experts from the *Confederación Hidrográfica del Júcar* (CHJ). Our results show that the automated variable selection outcomes align with CHJ's 25 years-long empirical refinement. In addition, the resultant FRIDA index outperforms the official State Index in terms of accuracy in reproducing the target variable and cardinality of the selected inputs' set.

## 1 Introduction

A drought is a slowly-developing natural phenomenon that occurs in all climatic zones and can be defined as a temporary significant decrease of water availability (Tallaksen and Van Lanen, 2004; Van Loon and Van Lanen, 2012). Drought impacts can propagate to virtually every water-related sector, such as farming and livestock production, industry, power generation, and public water supply (Spinoni et al., 2016). During the period 1976-2006, droughts in Europe affected more than 11% of the population, and their economic cost was estimated to exceed €100 billion, considering damages endured by consumers, tourism, industry, energy, and agricultural sectors. Moreover, climate change is expected to produce longer, more frequent and severe drought events, especially in southern Europe (Giorgi and Lionello, 2008; Spinoni et al., 2016; Marcos-Garcia et al., 2017). Recent drought cost trends show a significant increasing tendency, reaching an average of €6.2 billion/year in the years

1991-2006 (EU, 2007). These estimates, however, only account for the economic damages, (i.e., situations in which a water deficit induced by droughts affects production, sales and business in a variety of sectors), neglecting environmental and social costs (Spinoni et al., 2016). A comprehensive quantification of drought impacts is, in fact, complicated by the considerable lag occurring between the realization of dry climatic conditions and the impacts on economy and society (Changnon, 1987; Stahl et al., 2016).

We can distinguish four types of droughts: meteorological, agricultural, hydrological, and operational (or anthropogenic) drought, depending on the time horizon and the variable of interest. (Heim Jr, 2002; Mishra and Singh, 2010; Pedro-Monzonìs et al., 2015; Spinoni et al., 2016). The development chain of droughts through time is exemplified in Figure 1.

A meteorological drought is defined as a lack of precipitation over a region for a certain period of time (Mishra and Singh, 2010). It develops over the short term (1-3 months) and can extend on longer periods, and is usually associated with the global
behavior of the atmospheric circulation (Pedro-Monzonìs et al., 2015). Precipitation is always the core variable to characterize this drought type, with most meteorological drought indexes based on precipitation only (Byun and Wilhite, 1999; McKee et al., 1993). In some cases, especially in regions where droughts can be strongly influenced by evapotranspiration, additional variables such as temperature trends are also considered (Vicente-Serrano et al., 2010; Lorenzo-Lacruz et al., 2010).

Agricultural drought affects, and is defined through, the state of soils and crops in the medium term (3-6 months) (Pedro-
Monzonìs et al., 2015). This drought type manifests itself with dryness in the root zone and, although rainfall deficiency is a primary cause, precipitation alone is often not enough to describe it. Approaches to characterize agricultural droughts focus on monitoring soil water balance and the subsequent deficit (Palmer, 1965; Narasimhan and Srinivasan, 2005; Hao and AghaKouchak, 2013). The factors involved in this case include vegetation type, soil water holding capacity, wind intensity, evapotranspiration rate, and air humidity (Heim Jr, 2002). In regulated systems, agricultural droughts can be usually restrained
with irrigation (Keyantash and Dracup, 2002).

Hydrological drought is defined as a period of exceptionally low flows in watercourses, and lakes and groundwater levels below normal (Dracup et al., 1980; Van Loon and Van Lanen, 2012). Related indicators mainly focus on streamflow, as the by-product of every hydro-meteorological process taking place in water catchments (Heim Jr, 2002; Vicente-Serrano and López-Moreno, 2005). More comprehensive indexes can also include snowpack extent, reservoir storage, and groundwater level (Shafer and Dezman, 1982; Keyantash and Dracup, 2004; Staudinger et al., 2014). This drought takes place after a prolonged time of low precipitation and deficient soil moisture and its effects are witnessed in the long-term (6-12 months) (Zargar et al., 2011).

These three categories refer to droughts as a natural hazard, i.e., a threat of a naturally occurring event that negatively effects
5  people or the environment (Gustard et al., 2009; Van Loon and Van Lanen, 2013; Laaha et al., 2016). On the other hand, particularly in highly regulated contexts, a dry spell may be caused by natural scarcity of precipitation as well as inconsiderate overuse and/or mismanagement of water resources. Another interesting way to approach drought analysis is, therefore, through the concept of operational (or anthropogenic) drought. Operational drought is defined as a period with anomalous supply failures in a developed water system (Pedro-Monzonìs et al., 2015). It is caused by a combination of two factors: lack of
10  water resources and excess of demand (Mishra and Singh, 2010; AghaKouchak, 2015a). Moreover, it can be further worsened

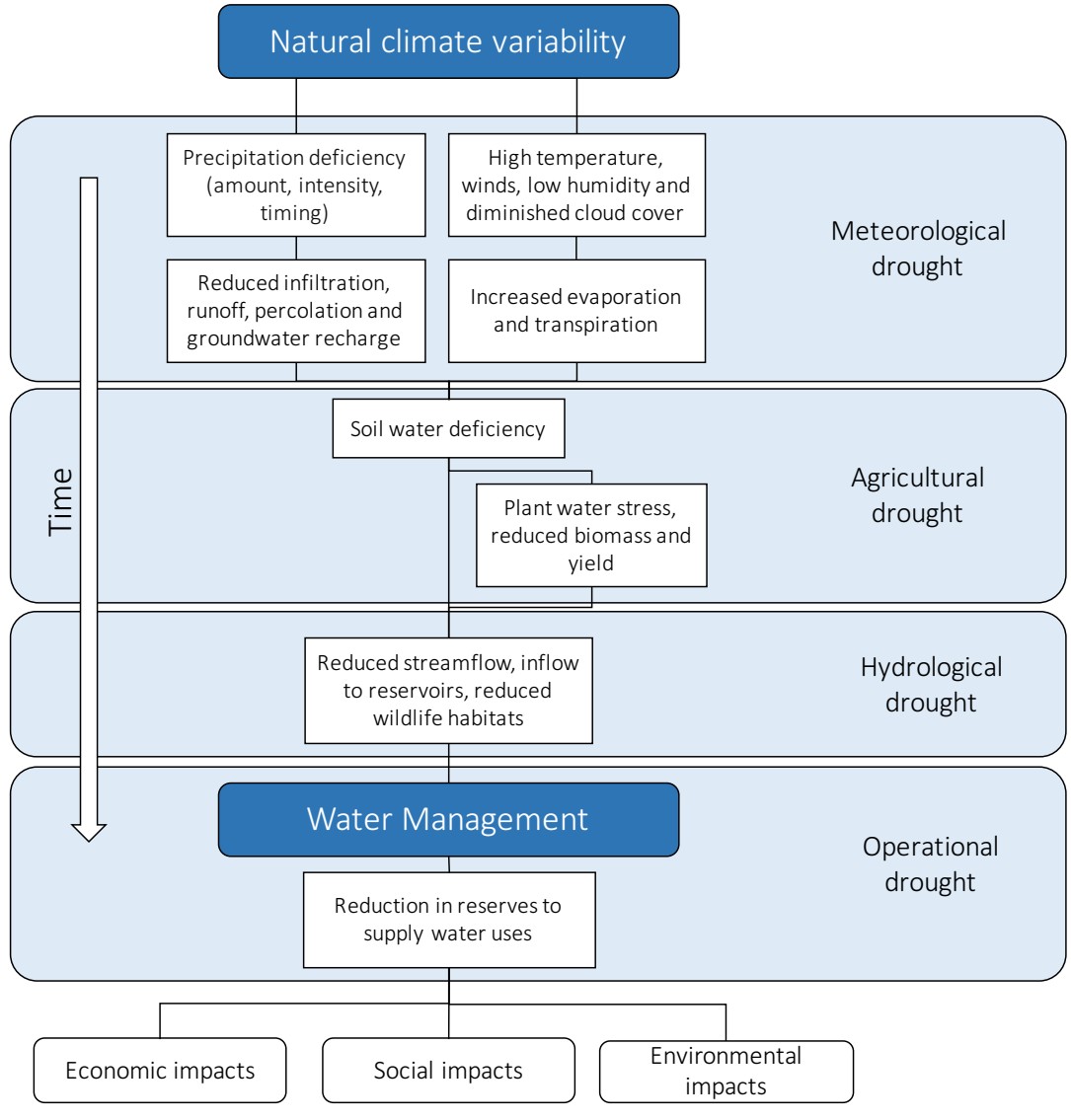

**Figure 1.** Development chain of droughts through time. Meteorological drought, defined as a lack of precipitation over a region for a certain period of time, develops in the short term. Agricultural drought accounts for the plants and crops water stress; develops in the medium term. Hydrological drought, defined as a period of low streamflow in watercourses, lakes and groundwater level below normal, develops in the long term. Operational drought, defined as a period with anomalous supply failures in a developed water exploitation system, develops in the long term. Figure adapted from Spinoni et al. (2016) to include Operational drought.

by an inadequate design and management of the water exploitation system and its operating rules (Mishra and Singh, 2010). Operational droughts indicators aim at comparing water availability to human water needs and serve as a measure of water well-being, rather than a measure of natural fluctuation as in the case of meteorological, agricultural, and hydrological indicators (Sullivan et al., 2003; Rijsberman, 2006). In the computation of operational drought indicators, the available water is often represented by the streamflow, or a fraction of it, and the water need is usually quantified by a standard per capita or by a fixed nominal demand (Falkenmark et al., 1989; Raskin et al., 1997). Depending on the application scope, operational drought indicators are either river basin specific (Garrote et al., 2007; Haro-Monteagudo et al., 2017) or used in studies covering continental or global areas with an annual time resolution (Yang et al., 2003; Oki and Kanae, 2006; Alcamo et al., 2007; Kummu et al., 2010).

When considering a highly regulated water system, i.e., a system where natural water availability is altered by the presence and operation of water infrastructures, traditional drought indicators (e.g., SPI, Standardized Precipitation Index; SPEI, Standardized Precipitation and Evapotranspiration Index; SRI, Standardized Runoff Index) present different shortcomings. On the one hand, meteorological, agricultural, and hydrological indexes often fail in representing drought conditions when regulated lake releases and/or groundwater pumping filter water availability and play a role in magnifying or smoothing drought impacts. Anthropized systems have, in fact, a demonstrated ability to endure meteorological droughts for months, or even years, without suffering consequences, i.e., without incurring in a situation of water shortage perceived by the users. An effective planning and management of water resources enables such systems to wisely exploit the combined storage capacities of surface and groundwater reserves and restrain drought (Rijsberman, 2006; Haro et al., 2014a). On the other hand, operational drought indexes are often designed to operate analysis over coarse spatiotemporal resolutions, thus resulting unsuitable for a real time basin level drought detection, characterization, and management. Highly regulated systems need ad hoc index formulations tailored on basin characteristics (Wanders et al., 2010; AghaKouchak, 2015b), combining human-controlled variables (e.g., reservoirs and groundwater levels) with uncontrolled hydro-meteorological variables (e.g., precipitation, temperature, natural inflows) to reflect both regulation effects and natural fluctuations in the basin.

A paradigmatic example of a practical and systematic policy for the identification and mitigation of operational droughts is provided by Spain, where public River Basin Management authorities (Confederaciones Hidrográficas) are bind by Law (Ministerio del Medio Ambiente, 2000) to design basin-specific State Indexes associated with each main river basin ($Ie$, *Índice de Estado*). Most of the basins in Spain are highly regulated and these State Indexes are computed as a weighted average of relevant observed variables at selected control points, e.g., precipitation, streamflow, reservoir level, and groundwater level. Each river basin authority has designed its customized formulation for the State Index which reflects the hydroclimatic conditions and the water uses of the region (Estrela and Vargas, 2012). The value of the State Indexes is monitored monthly and used to trigger water demand and supply measures when entering a drought period, according to the district Drought Management Plan (DMP) (Garrote et al., 2007; Gómez and Blanco, 2012; Haro et al., 2014a).

Each DMP and the relative State Index formulation is the result of a long collaborative process including public participation, basin experts, and stakeholders, and providing an effective multi-sector partnership approach for managing drought risk (Carmona et al., 2017). State Indexes are the result of a long trial-and-error process mostly begun in the eighties, through which

the variable choice and combination have been progressively adjusted to best suit the basin drought management requirements. In the case of the Jucar basin, for instance, the final form of the associated index was established in 2007 with a report by the *Confederación Hidrográfica del Júcar* (CHJ, 2007a), after 25 years of refinements. This long empirical process produced an index formulation tailored for the Jucar system, which cannot be generalized to different contexts. Similarly, other main Spanish river basins (e.g., Duero, Ebro, and Guadalquivir river basins) underwent an analogous process and formulated their own State Indexes (CHD, 2007; CHE, 2007; CHG, 2007).

Since their establishment in 2007, State Indexes have represented the most consistent and extensively applied paradigm of index-based drought management. Thus, $I_e$s constitute the state of the art for basin-customized operational drought indexes. A reasonable research question is whether the empirical process leading to their design can be formalized, automated, and easily exported to different water systems.

In this study, we contribute the FRamework for Index-based Drought Analysis (FRIDA), which allows the automatic construction of basin-customized drought indexes for highly regulated water systems. In contrast to traditional empirical approaches, FRIDA uses an advanced feature extraction method that completely automatizes and generalizes the variable selection process for the construction of the index. The selected variables are then combined into a new index that can effectively represent the state of water resources in the basin as well as support the characterization of drought conditions. The feature extraction step is key in FRIDA as it guides the construction of a skillful (highly accurate) and parsimonious (with low input dimensionality) drought index by performing the selection of the best input subset to build a model of a predefined target output representing the drought conditions in the basin.

Specifically, FRIDA is structured in three steps. First, we define a target variable, an appropriately chosen water deficit acting as a proxy for the drought conditions of the considered basin (e.g., water supply deficit, soil moisture deficit), and a dataset of hydro-meteorological variables and traditional drought indicators. Second, we identify Pareto optimal subsets of variables balancing predictive accuracy and parsimony. In this study, we employed the Wrapper for Quasi-Equally Informative Subset Selection (W-QEISS) to perform this operation (Karakaya et al., 2015; Taormina et al., 2016). Traditional variable selection algorithms are conceived to select only one optimal subset of predictors, while W-QEISS identifies one subset with the highest predictive accuracy, and multiple subsets with similar information content, thus providing more informative results. Moreover, W-QEISS includes two metrics of relevance and redundancy in the search process in addition to the commonly used objectives of accuracy and cardinality, fostering the diversification among the provided solutions (Sharma and Mehrotra, 2014). Third, we choose the preferred predictor subset among the non-dominated solutions based on accuracy, cardinality (i.e., dimensionality), and, possibly, additional factors, including cost and availability of the variable observations. The subset is finally used to calibrate a chosen model class with respect to the target variable, and the drought index is thus completed.

The potential of the proposed framework is demonstrated on the highly regulated Mediterranean basin of the Jucar river, in eastern Spain, where the State Index-based drought management system provides an ideal benchmark for testing FRIDA index (Andreu et al., 2009; Haro et al., 2014b; Pedro-Monzonís et al., 2014; Macian-Sorribes and Pulido-Velazquez, 2017; Haro-Monteagudo et al., 2017; Carmona et al., 2017). The Jucar State Index provides guidelines for FRIDA application. First, it facilitates the target variable choice and candidate variable retrieval, and, second, it allows the validation of FRIDA

predictors selection, and index design steps. FRIDA and State indexes are compared in terms of accuracy in reproducing the drought conditions of the basin, number of variables required for their computation, and general reliability and portability of the methods. The outcome of this analysis consists in demonstrating the validity of a completely automated procedure (i.e., no information on system topology or basin characteristics is required) in recognizing the main drought drivers, and predicting a deficit with accuracy and limited computational effort.

## 2 Methods and tools

### 2.1 Framework for Index-based Drought Analysis

The FRamework for Index-based Drought Analysis (FRIDA) designs drought indexes in three steps as reported in Figure 2.

The Identification of basin characteristics is a preliminary empirical process, which consists in the selection of a target variable and the collection of candidate predictors. The target variable is an appropriately chosen water deficit, representative of the actual drought conditions in the basin (e.g., water supply deficit, soil moisture deficit). The dataset of predictors contains the candidate features to reproduce the target variable and consists of observed hydro-meteorological variables and composite drought indicators over different spatio-temporal scales.

Target variable and candidate predictors constitute the input to the Feature Extraction step, the second building block of the framework. This block employs an Input Variable Selection (IVS) algorithm that explores the space of candidate predictors to select Pareto efficient subsets of predictors with respect to multiple assessment metrics. Most commonly, these metrics quantify the subset accuracy in reproducing the target and the parsimony (i.e. the cardinality of the subset), crucial characteristics for an operational index expected to balance precision and ease-of-use. In some cases, also relevance and redundancy can be considered in order to explore the input space more effectively. In particular, the metric of relevance favors highly informative subsets (i.e., constituted by predictors that are highly correlated with the target), while the redundancy metric ensures low intra-subset similarity. The objectives of relevance and redundancy are essential to stimulate the search process towards the identification of a diversified and comprehensive set of solutions, which would not be achieved optimizing cardinality and accuracy only.

In this work, we use an advanced IVS algorithm called Wrapper for Quasi-Equally Informative Subset Selection (W-QEISS). W-QEISS provides as output a number of efficient subsets that are collected in a Selection Matrix.

In the Drought Index modeling block, the preferred efficient solution is selected by the user, balancing the trade-off between competing objectives, and, possibly, considering additional operative needs neglected in the IVS search (e.g., cost and reliability of the variable monitoring). Lastly, an appropriate regressor is fit to the sample data set of Pareto efficient inputs and the target variable. The choice of model class is determined by the application of interest. In general, highly non-linear learning machines like Artificial Neural Networks (ANNs) provide a good balance between accuracy and flexibility. On the other hand, such black-box models lack of intuitive interpretability and might result unsuitable for applications that affect several stakeholders and require a wide acceptance of the tool to be employed (Estrela and Vargas, 2012). In these cases, a simpler model (e.g., a linear

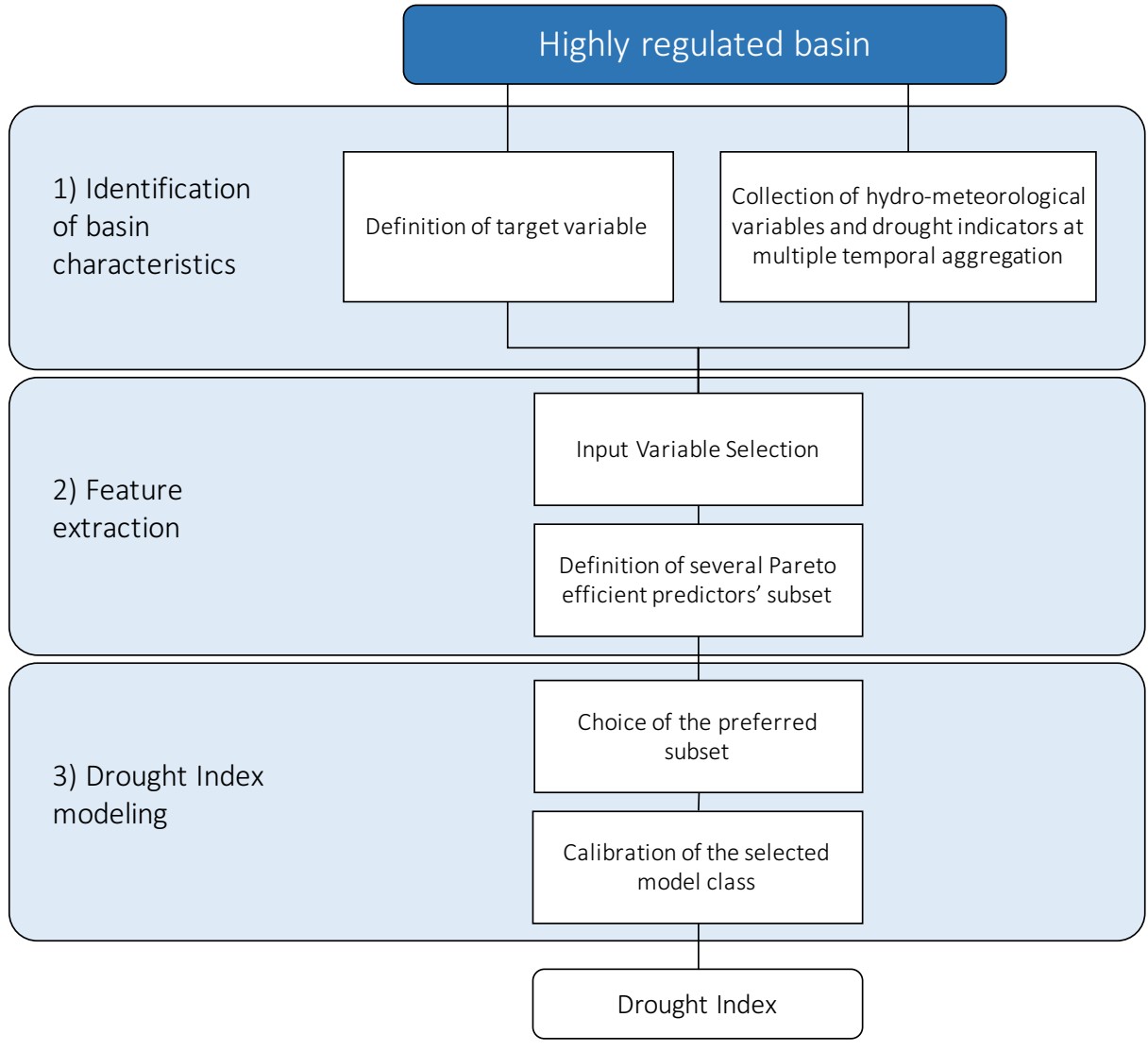

**Figure 2.** FRamework for Index-based Drought Analysis (FRIDA): 1. Identification of basin characteristics, 2. Feature Extraction, 3. Drought Index modeling.

model) might be preferred, as it grants an immediate understanding of the physical meaning, though at the price of poorer approximation skills.

## 2.2 Feature Extraction via Wrapper for Quasi-Equally Informative Subset Selection

Feature extraction techniques, employed in the second block of FRIDA, are an ensemble of data pre-processing algorithms that transform the original input data set into a more compact, while still highly informative, subset (Cunningham, 2008). Among

15 the feature extraction algorithms, Input Variable Selection (IVS) techniques specifically address the problem of the reduction of the input space by identifying the relevant predictors to be used to calibrate a model of the target variable (Bowden et al., 2005). There are two main classes of IVS techniques: Filters and Wrappers. Filters evaluate the relevance of each variable separately, computing an error metric on the features (Yang and Pedersen, 1997; Sharma, 2000; Galelli and Castelletti, 2013). Wrappers, on the other hand, assess the relevance of a variables ensemble, evaluating the prediction performance of a given learning

machine calibrated on the input set, and thus considering the interactions and dependencies between variables (Guyon, 2003). In terms of performance, Wrappers are often more accurate than Filters, although computationally more intensive (Galelli et al., 2014).

In this study, we used the Wrapper for Quasi-Equally Informative Subset Selection (Karakaya et al., 2015; Taormina et al., 2016). The W-QEISS algorithm receives as input the set $\mathbf{X}$ of candidate predictors, i.e., $\mathbf{X} = \{x_i, \ldots, x_{n_X}\}$ and the trajectory

$y$ of the target variable. The algorithm is composed of three main steps (Karakaya et al., 2015), as synthesized in Figure 3:

- Step 1: a set $\mathbf{A} \subseteq \mathbf{X}$ of Pareto-efficient solutions is built according to the four-objective functions of relevance $f_1(\cdot)$, redundancy $f_2(\cdot)$, cardinality $f_3(\cdot)$, and accuracy $f_4(\cdot)$. A global multi-objective optimization algorithm is employed to explore the space of the possible subsets. In this study, we use the self-adaptive Borg MOEA (Hadka and Reed, 2013), which has shown to outperform other benchmark evolutionary algorithms in terms of number of solutions returned,

ability to handle many-objective problems, ease-of-use, and overall consistency across a suite of challenging multi-objective problems (Reed et al., 2013). A learning machine is used to compute the predictive accuracy $f_4$ of each set. In this study, we employ the Extreme Learning Machines (ELMs) (Huang et al., 2006), belonging to the family of Artificial Neural Networks, which were shown to provide a good performance in terms of accuracy and flexibility in a variety of problems while resulting up to thousand times faster than benchmark feedforward ANNs (Huang et al., 2012). ELMs, in fact, bypass the time consuming gradient-based search of optimal neurons parameters required by traditional ANN techniques, by defining randomly parameterized hidden nodes, and subsequently optimizing their output weights. Such optimization is solved through a one-step matrix product and essentially amounts to learning a linear model.

5 However, we do not expect the choice of the learning machine or MOEA to be crucial for the attainment of the result. A different benchmark MOEA (e.g., NGSAII, MOEAD, eps-MOEA) is likely to achieve a comparable result, although requiring a possibly significant effort in the manual calibration of the evolution parameters, which is automated in Borg MOEA. Similarly, other ANN techniques could in principle be substituted to ELM, although incrementing the

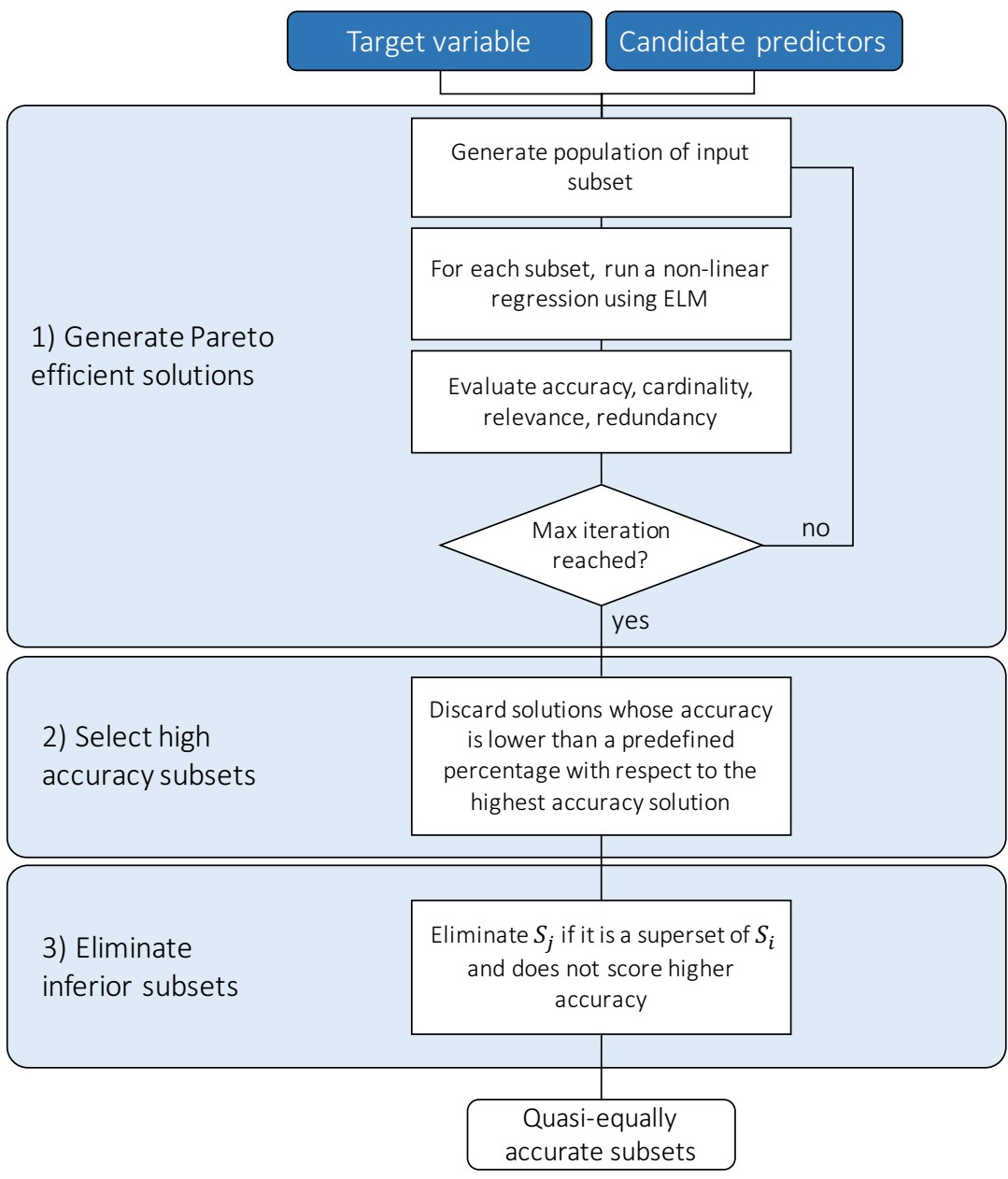

**Figure 3.** W-QEISS flowchart. Step 1: generate Pareto efficient solutions with respect to the four objectives of relevance, redundancy, cardinality, and accuracy; Step 2: select high accuracy subsets; Step 3: eliminate inferior subsets.

computational time to possibly unbearable levels, given the multiple calibration and validation processes reiterated in WQEISS.

- Step 2: Among the Pareto-efficient subsets, the maximum value of accuracy $f_4^*$ is identified, associated with subset $\mathbf{S}_{f_4^*} \subseteq \mathbf{A}$. Then, solutions with significantly lower accuracy are discarded and from ensemble $\mathbf{A}$, obtaining $\mathbf{A}_\delta$. The ensemble $\mathbf{A}_\delta$ contains quasi-equally informative subsets with respect to $\mathbf{S}_{f_4^*} \subseteq \mathbf{A}_\delta \subseteq \mathbf{A}$, i.e., subsets that have (almost) the same predictive accuracy with respect to a given model class. When the dataset of candidate variables presents significant correlation among features, numerous subsets characterized by a wide range of cardinalities are generally available to achieve a relative small range of accuracies. This is often the case in environmental problems, where spatial and temporal correlation of hydro-meteorological variables and associated indicators is significant. Therefore, at this stage, the accuracy metric is used to retain accurate solutions only, provided that they feature different cardinalities and predictors combinations.

  Formally, on the basis of an predefined small value of $\delta$, $\mathbf{S}_i$ is $\delta$-quasi equally informative to subset $\mathbf{S}_{f_4^*}$ if

$$f_4(\mathbf{S}_i) \geq (1-\delta)f_4^* \qquad for \quad 0 \leq \delta \leq 1 \tag{1}$$

- Step 3: The final ensemble $\mathbf{A}_\delta^*$ is computed after the elimination of the inferior subsets. The subset $\mathbf{S}_j$ is considered inferior to $\mathbf{S}_i$, if it is a superset of $\mathbf{S}_i$, and does not score higher accuracy. Formally
  $\mathbf{S}_i \subset \mathbf{S}_j$ and $f_4(\mathbf{S}_i) \geq f_4(\mathbf{S}_j)$.

  In this step, all subsets contained in $\mathbf{A}_\delta$ are compared in order to find possible inferior subsets and eliminate them. By doing this, the final ensemble of $\delta$-quasi equally informative subsets $\mathbf{A}_\delta^*$ is provided as output of the procedure and reported in a Selection Matrix.

The W-QEISS algorithm differs from a traditional IVS approach as it introduces the consideration that, for a given cardinality, multiple subsets of variables can have almost indistinguishable accuracy performance. The outcome of W-QEISS variable selection is thus not a single most accurate subset for each cardinality, but a pool of $\delta$-quasi equally accurate solutions among which the preference can be determined by other metrics not directly considered in the optimization (e.g., cost and reliability of the variable observation).

Another innovative feature of the W-QEISS approach relies on the formulation of a four objective optimization problem. Beside the two traditional objectives of accuracy ad complexity commonly employed in Wrappers, W-QEISS includes other two metrics of relevance and redundancy (Sharma and Mehrotra, 2014). The maximization of accuracy ensures a precise reproduction of the data, while the minimization of cardinality aims at simplifying the final models. These characteristics are key for an operational index, expected to balance precision and ease-of-use. Relevance and redundancy optimization is instead an asset for an effective subset search process, as it fosters the diversification of the solutions explored within the MOEA algorithm, guaranteeing low intra-subset similarity, and high information content of the solutions. A two-objective search based on cardinality and accuracy only would, in fact, identify optimal solutions, but at the same time disregard a number of quasi-equally informative subsets with an almost identical operational behavior. The identification of such alternative solutions,

nevertheless, grants flexibility and a multiplicity of options for the expert-based choice of the preferred subset, where certain combinations of predictors can be favored according to case-specific operative purposes, e.g., more robust or less costly data gathering process, enhanced acceptability or immediacy of the index.

Three of the four objectives formulations make use of the Symmetric Uncertainty (SU), a measure of the dependence and similarity between two variables (Witten and Frank, 2005). SU assumes values between 0 (independent variables) and 1 (complete dependence) and is computed for two features $A$ and $B$ as:

$$SU(A,B) = \left[ \frac{2 \cdot (H(A) + H(B) - H(A,B))}{H(A) + H(B)} \right] \tag{2}$$

where $H(\cdot)$ is the entropy of variable $(\cdot)$ (see for instance Scott (2012) for the definition).

WQEISS bases its objectives formulation on information theory, as discussed in Karakaya et al. (2015). Information theoretic criteria (e.g., SU, Mutual information, and Partial Mutual Information) do not assume any functional relationship between the variables and thus result well suited to quantify the dependence between two variables in any modeling context (MacKay, 2003). Other objectives formulations could in principle be explored, for instance substituting the use of Symmetric Uncertainty with more traditional correlation coefficients, although with the risk of losing generality by assuming linear dependence between variables.

The four assessment metrics are formulated as follows:

1. Relevance $f_1(\mathbf{S})$: to be maximized, is formulated as:

$$f_1(\mathbf{S}) = \sum_{x_i \in \mathbf{S} \subseteq \mathbf{X}} SU(x_i, y) \tag{3}$$

where the term $SU(x_i, y)$ represents the symmetric uncertainty between the feature $x_i$ and the output $y$. The relevance is therefore a measure of the explanatory power of the features with respect to the output.

2. Redundancy $f_2(S)$: to be minimized, is formulated as:

$$f_2(\mathbf{S}) = \sum_{x_i \in \mathbf{S} \subseteq \mathbf{X}} SU(x_i, x_j) \tag{4}$$

where $SU(x_i, x_j)$ represents the SU between two features $x_i$ and $x_j$. High redundancy thus means high similarity between the features. By minimizing the redundancy the algorithm ensures that the search will be oriented towards the selection of subsets with mutually dissimilar features.

3. Cardinality $f_3(\mathbf{S})$: to be minimized, is formulated as:

$$f_3(\mathbf{S}) = |\mathbf{S}| \tag{5}$$

where $|\mathbf{S}|$ is the number of predictors within the subset. Its minimization guarantees that the resulting model will not be unnecessarily complex.

4. Accuracy $f_4(\mathbf{S})$: to be maximized, is formulated as:

$$f_4(\mathbf{S}) = SU(y, \widehat{y}(\mathbf{S})) \tag{6}$$

where $SU(y, \widehat{y}(\mathbf{S}))$ is the correlation, measured in SU, between the observed output $y$ and the prediction $\widehat{y}(\mathbf{S})$ obtained from the model.

## 3 Case Study: the Jucar river basin

The Jucar river basin occupies an area of 42,989 km$^2$ located in the eastern part of Spain (see Figure 4). The territory is mainly mountainous in the interior part, while the center-eastern section shows a vast plain system ending into the Mediterranean sea. The territory is characterized by various climatic conditions of which sub-humid and semi-arid are largely dominating. The main rivers of the area are Jucar, Mijares, and Turia, covering all together more than 80% of the total mean areal flow. The subterranean runoff is very relevant, providing 74% of the contribution to the river network (CHJ, 2007a).

Since the mean value of the total annual runoff (1,747 Mm$^3$ from 1940 to 2009) almost equals the annual water demand (1,640 Mm$^3$), water scarcity and droughts have long been perceived as primary issues for agricultural, social, economic, and environmental reasons. On the other hand, meteorological droughts in the Jucar basin can be endured for several years without suffering any consequences, due to the highly regulated water system set in the area. There are three main large surface reservoirs in the region: Alarcón, Contreras, and Tous (maximum capacity: 1,118 Mm$^3$, 444 Mm$^3$, and 378.6 Mm$^3$, respectively). In addition, most aquifers in the basin are intensively exploited to support agricultural supply and are currently experiencing a significant depletion due to over-drafting, which, in turn, affects the rivers flow.

In such a highly regulated basin with long overyear storage, water scarcity is not a necessary condition derived from a meteorological drought (CHJ, 2007a; Carmona et al., 2017). Thus, traditional drought indexes fail in detecting the timing and severity of the incidence of a drought, and an ad-hoc monitoring system was conceived to properly capture the hydrological status of the catchment. The monitoring system is based on the formulation of a basin specific index, namely the State Index ($\mathbf{I}_e$, *Índice de Estado*). The State Index was constructed empirically by the Jucar river basin authority (CHJ), with the intent of highly correlate to water scarcity conditions in the basin, in order to support drought management and the implementation of the actions considered in the Drought Management Plan (CHJ, 2007a). For that purposes, the index is developed after identifying the water sources for every main demand in the basin and the selection of representative variables to characterize the status of those sources.

The total State Index $\mathbf{I_e}$ is computed as a weighted mean of 12 partial $\mathrm{I}_e$. Partial $\mathrm{I}_e$s are obtained by normalizing hydro-meteorological indicators ($\mathrm{V}_i$) belonging to the following categories (see Figure 4):

1. The mean monthly storage of one, or more reservoirs combined [Mm$^3$] (2 storage *indicators*);

2. The mean streamflow contribution of the last 3 months [Mm$^3$] (4 flow *indicators*);

3. The mean monthly piezometric level [m] (3 piezometer *indicators*);

4. The areal precipitation of the last 12 months [mm], computed averaging the values observed by multiple pluviometers (3 precipitation *indicators*).

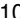

**Figure 4.** Map of the Jucar Basin river network. The colored markers represent the variables considered for the State Index calculation. S: reservoir storage, F: streamflow, Pz: piezometer, Pl: pluviometer. Streamflow and piezometers markers are located in correspondence to the relative measurement station, while storage and pluviometers markers are put in the center of the polygon formed by connecting the multiple measurement points used for their computation.

Each *indicator* (*Vi*) is consequently normalized to obtain 12 partial $I_e$ values:

$$I_e = \begin{cases} \dfrac{1}{2}\left[1 + \dfrac{Vi - Vm}{Vmax - Vm}\right] & \text{if } Vi \geq Vm \quad (7a) \\ \dfrac{Vi - Vmin}{2\,(Vm - Vmin)} & \text{if } Vi < Vm \quad (7b) \end{cases}$$

where *Vm*, *Vmax* and *Vmin* are the mean, maximum, and minimum values of each indicator time series. The storage and precipitation monthly time series are normalized with respect to maximum and minimum values of the considered month, while piezometers and river flows are normalized with respect to the complete historical time series. The partial $I_e$s result as normalized indexes between 0 and 1, where $I_e > 0.5$ indicate higher than average value of *Vi*. Once the partial $I_e$ have been computed, they are aggregated as a weighted sum to obtain the total $\mathbf{I_e}$. The weights are established according to the demand class associated to the indicator, ranging from class A (demand > 100 hm$^3$/year) to D (demand < 10 hm$^3$/year).

The Jucar river basin represents a Mediterranean drought prone highly regulated basin, featuring one of the most innovative and effective drought management systems, relying on the formulation of an empirically constructed basin specific drought index (Andreu et al., 2009; Haro et al., 2014b; Haro-Monteagudo et al., 2017; Carmona et al., 2017). As a consequence, it represents the state of the art for basin-customized operational drought indexes employed for drought restraining purposes, and a remarkable benchmark to test and validate the proposed FRIDA methodology.

## 4 Numerical results

For the presentation of the numerical results we follow the workflow proposed in Figure 2 . The length of the dataset available for the experiments is $N = 174$ data points, corresponding to monthly values in the period 1986-2000, and $n_x = 28$ number of candidate predictors were used (Zaniolo et al., 2018). The parameterization of W-QEISS was adjusted using available guidelines given by Huang et al. (2006), Karakaya et al. (2015), and a trial-and-error process. For Borg MOEA, we set the number of function evaluation (NFE) equal to 2 millions, while the number of hidden neurons in the ELM, presenting a sigmoidal activation function, was set to 30. A k-fold cross-validation process (with k = 10) was repeated 5 times and the average resulting value was used to estimate the predictive accuracy of each model. The W-QEISS experiment with such setting was run 20 times to filter out the random component of the process, and the results presented below are obtained by merging the Pareto fronts obtained by each repetitions into a final Pareto front of non-dominated solutions.

### 4.1 Identification of basin's characteristics

In the first report concerning the $\mathbf{I_e}$ development (CHJ, 2007b), the index was validated for the time span from January 1986 to June 2000 against the supply deficit recorded in the basin with respect to agricultural and urban water demand, and the procedure for the State Index computation was approved. To ensure comparability between the $\mathbf{I_e}$ and the FRIDA constructed index, we decided to employ the same supply deficit as target variable for the application of FRIDA approach to the Jucar case study. The Jucar supply deficit employed in this work was simulated via AQUATOOL model (Andreu et al., 1996). The model can run in simulation mode with a monthly time step, and it is conceived in the form of a flow network with oriented connections reproducing water losses, hydraulic relations between nodes, reservoirs and aquifers, and flow limitations based on elevation. Within AQUATOOL, complex processes such as evaporation and infiltration are effectively reproduced. The modeled supply deficit, employed as target variable, represents the monthly nominal shortage of water conveyed to the irrigation districts, and is only quantifiable a posteriori, when the water shortage has already jeopardized the fields. On the other hand, a drought index

can be constantly monitored, and thus represents a valuable management tool for restraining drought impacts and identifying effective drought management strategies.

20    The database of candidate input variables was assembled retrieving the available observed variables in the basin and computing traditional drought indicators at multiple time aggregations. The resulting candidate predictors, listed in Table 1, are the following:

- 2 temporal features: date of the measurement, and month of the year;

- 12 monthly observed variables, current inputs to the $\mathbf{I_e}$, reported in Figure 4: average monthly storage and groundwater levels, average three months river runoff, and cumulated areal precipitation over 12 months;

- 8 additional observed variables in the basin: outflows from, and inflows to, the main reservoirs, and mean monthly areal temperatures;

- 6 traditional drought indicators: Standardized Precipitation Index (SPI), and Standardized Precipitation and Evapotranspiration Index (SPEI). SPI and SPEI indicators are computed on mean monthly data over the entire basin for 3, 6, and 12 months time aggregations. SPI requires as input the precipitation, and SPEI requires precipitation and temperature, as it uses the difference between precipitation and potential ET as reference variable.

Their values express the water availability conditions of a basin in terms of units of standard deviation from the mean: negative (positive) values indicate drier (wetter) conditions than average (see McKee et al. (1993); Vicente-Serrano et al. (2010) for details on definition and calculation of these indicators).

## 4.2   Feature extraction via W-QEISS

The result of the W-QEISS algorithm is not a single most-accurate set of variables for a given cardinality, but several quasi-equally informative subsets, whose accuracy is lower than the best one by a small percentage $\delta \cdot 100\%$. Figure 5 represents a Selection Matrix, which reports the composition of each alternative subset of predictors within 15% of accuracy with respect to the highest one. The value $\delta = 0.15$ was chosen since it provides a reasonable trade-off between the number of solutions and their accuracy. The accuracy is measured in symmetric uncertainty between the target variable and the ELM calibrated using the reported subset.

The alternative subsets are sorted in ascending order of cardinality (from top to bottom), and accuracy (within each cardinality level). A rectangular marker is placed at the intersection between the row identifying a given subset and the columns corresponding to the selected predictors. The marker color varies with the cardinality of the subset, with lighter shades of gray indicating smaller subsets. In this case the cardinality spans from 3 to 9 features. The highest accuracy is reported in red and recorded for subset number 14. The 5 corresponding selected predictors, marked on the horizontal axis with a blue background, are the following:

**Table 1.** Set of candidate input features for the feature extraction step via W-QEISS.

| Feature type | Feature code | Description |
|---|---|---|
| Time information | Date | Date of the measurement |
| | Moy | Month of the year |
| State Index Inputs | S1 | Cumulated storage of Alarcón, Contreras and Tous |
| | S2 | Storage at Forata |
| | F1 | Flow measurement in the upper basin |
| | F2 | Flow measurement in the upper basin |
| | F3 | Flow measurement in the middle basin |
| | F4 | Flow at Jardín tributary |
| | Pl1 | Pluviometer measurement in Contreras reservoir |
| | Pl2 | Pluviometer measurement in Tous reservoir |
| | Pl3 | Pluviometer measurement in Bellús reservoir |
| | Pz1 | Piezometric level in the south-east |
| | Pz2 | Piezometric level in the center |
| | Pz3 | Piezometric level in the west |
| Observed variables | In A | Inflow to Alarcón reservoir |
| | In C | Inflow to Contreras reservoir |
| | In T | Inflow to Tous reservoir |
| | Out A | Outflow from Alarcón reservoir |
| | Out C | Outflow from Contreras reservoir |
| | T1 | Temperature in the west |
| | T2 | Temperature in the center |
| | T3 | Temperature in the east |
| Indicators | $SPI_3$ | SPI at 3 months time aggregation |
| | $SPEI_3$ | SPEI at 3 months time aggregation |
| | $SPI_6$ | SPI at 6 months time aggregation |
| | $SPEI_6$ | SPEI at 6 months time aggregation |
| | $SPI_{12}$ | SPI at 12 months time aggregation |
| | $SPEI_{12}$ | SPEI at 12 months time aggregation |

– Moy: month of the year;

15 – S1: total storage aggregated for the reservoirs Alarcón, Contreras, and Tous;

– F3: river flow measured in the Jucar middle basin, after the confluence with smaller rivers Jardín and Lezuza coming from south-west;

– Pz2: groundwater level measured at the Piezometer situated in central area of the basin, in correspondence of a rainfed agricultural area;

20 – $SPEI_6$: SPEI at 6 month time aggregation computed with precipitation and temperature data averaged for the whole basin.

From the analysis of the Selection Matrix, several insights can be gained from a modeling and from a decision-making viewpoints. To begin with, insights on predictors' relevance can be obtained from the detection of the vertical bars traced by joining markers across multiple rows. Uninterrupted bars indicate strongly relevant predictors that cannot be substituted by 25 other input combinations without incurring into a substantial drop of predictive accuracy. This is the case of the cumulated storage of the three main reservoirs Alarcón, Contreras, and Tous (S1). This information is essential to the final model, as the exclusion of such predictors highly affects the model performance. Increasing gaps in the vertical bars are found when considering predictors with progressively weaker relevance, while irrelevant inputs are recognizable by isolated markers or their total absence. The variables Moy, F3, and Pz2 are considered relevant variables, as they are selected quite frequently, 30 although high accuracy solutions exist that do not make use of all of them. Finally, the variable $SPEI_6$, while included in the most accurate subset, is overall present in 4 subsets only, whereas in other solutions with comparable accuracy it is replaced by different predictors, mainly carrying a similar precipitation-based information, such as pluviometer measures, or SPI, SPEI indicators at different time aggregations.

The presence of alternative subsets helps exploring the trade-off between multiple measures of predictive accuracy with respect to other metrics not directly considered in the optimization routine, an the choice of the preferred subset is determined 5 by the index application. Given the cardinality, one can decide to sacrifice a small amount of predictive accuracy for an easier-to-yield (or more reliable) combination of predictors. For example, with a loss smaller than 1% in accuracy, subset 13 selects $SPI_6$ instead of $SPEI_6$. This possible replacement is interesting from an point of view as SPI is easier to compute than SPEI. In fact, SPI requires only the precipitation for its computation with respect to precipitation and temperature or evapotranspiration needed for the computation of SPEI. In addition, even after the preferred subset is chosen and the system is 10 operating, knowing that one specific predictor can be replaced by one (or multiple) predictor(s) can aid the management in case of monitoring networks maintenance or instrument failure. When the main predictor is not observable, one can temporarily resort to alternative predictors incurring in a minimum loss of accuracy.

An additional consideration is related to the possibility to effectively address the uncertainty deriving from the choice of model inputs (Taormina et al., 2016). When multiple alternative subsets are provided, it is possible to explore the uncertainty 15 related to the selection of predictors yielding similar accuracy. For instance, in this case study, we can observe that almost

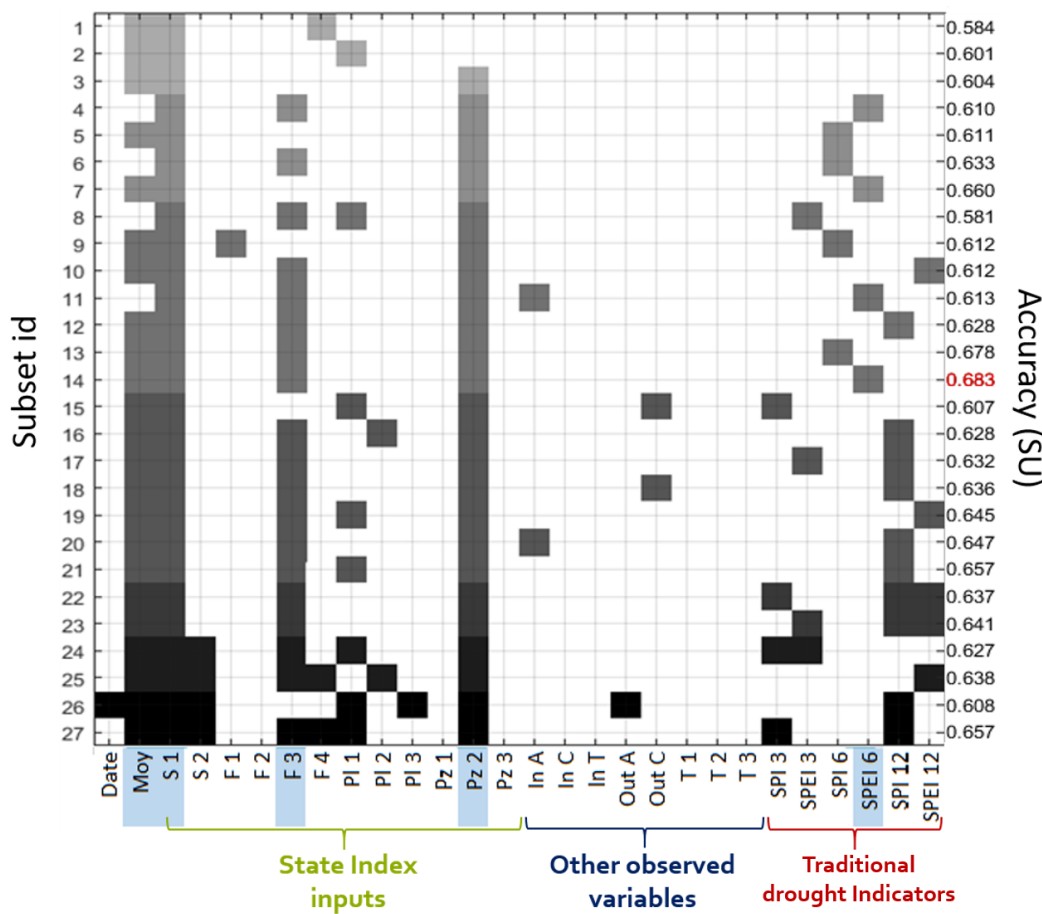

**Figure 5.** Selection Matrix: the left vertical axis represents the subset number and the right vertical axis the corresponding accuracy measured in SU. A colored marker is put in correspondence of the variables, listed on the horizontal axis, selected by each subset. The shade of gray is an indication of the cardinality of the subset, lighter shades for lower cardinality. The highest accuracy is reported in red and the corresponding variables, constituting the most accurate subset, have a blue background.

all subsets carry a groundwater and a rain information, but while the piezometric level is consistently provided by Pz2, the source of the precipitation information highly varies among the precipitation-based features (pluviometers or other SPI, SPEI indicators).

Finally, through the selection matrix analysis we can contrast the features selected by W-QEISS and the variables that constitute the State Index input set. Apart from sporadic single selections, all the observed variables not included in the State Index are consistently discarded by the W-QEISS as well, suggesting that the algorithm comes to the same conclusion as the Spanish experts considering inflows, outflows, and temperatures as non-relevant for the description of the state of water resources in the Jucar river basin. Note that this result is a consequence of the use of the nominal agricultural demand to compute the target deficit. A temperature information is likely to become relevant if a real, weather-influenced, agricultural demand is employed instead. The feature month of the year is not explicitly an input to the State Index, nevertheless, an analogous information is implicitly included in the $I_e$ through the normalization of the indicators described in equation 7. On the other hand, several features are considered in the $I_e$, but generally neglected by W-QEISS selection. Among them, two out of three piezometers, the river flows upstream from the reservoirs, one pluviometer and the storage of Forata. These inputs probably result redundant due to their spatial correlation. Spatial variability is considered in the computation of $I_e$ by including several spatially distributed observations of the main information categories: 2 measures of reservoir storages, 4 of river flows, 3 groundwater levels, and 3 precipitation measures. Conversely, the selection matrix supports the gain of a deeper understanding of the spatial interdependence of variables by identifying the best location for measuring the variables, spearing the need for several distributed measures. The highest accuracy-subset, in fact, selects only one variable out of each category: 1 storage, 1 river flows measure, 1 piezometer, and a spatially distributed precipitation information, i.e., $SPEI_6$ which replaces three areal pluviometers.

### 4.3 Drought Index Modeling

Among the pool of solutions, the choice of the preferred subsets is driven by the index application. For instance, an on-line use of the index that requires its frequent computation may benefit from an agile, easy-to-observe subset. With respect to the highest accuracy solution (subset 14), for instance, subset number 7 neglects predictor F3 thus presenting lower cardinality with an accuracy loss of only 3%. Similarly, the already mentioned subset 13 contains an easier-to-compute indicator (SPI instead of SPEI) with a negligible performance degradation. Nevertheless, for our methodological purpose we will employ the most accurate subset 14, as we are interested in discussing the potential of the method.

Concerning the model class choice, a highly flexible non-linear model is likely to yield the highest accuracy in reproducing the target. However, strong non-linearity and black-box behavior typically result in poor interpretability, a feature that is detrimental to the use of the index for management purposes as in the Jucar system, where restrictive measures in water use are activated when certain threshold values of the State Index are reached. As a consequence, the index outcome exerts a direct influence on many water-related activities requiring an easily interpretable and widely acceptable tool.

The calibration of a linear model on the chosen 5 dimensional subset seems to be a good compromise between accuracy and transparency. As mentioned above, the feature Moy represents the succession of the months in the year, and is an expression

of the seasonality of hydro-meteorological processes. Moy is constructed as the repetition of an array of numbers from 1 to 12 for the length of the considered time horizon, and thus presents a discontinuous shape: a slow and steady increase followed by a steep decrease in correspondence to the onset of a new year. While the non-linear models employed in the feature selection can effortlessly handle such an intermittent vector, linear models struggle with similar shapes. We therefore decided to account for the seasonality in the linear model indirectly, i.e., excluding Moy from the predictors set, but, consistently, considering seasonality by depurating the predictors of their annual cyclostationary mean.

The calibrated linear model representing the supply deficit is reported in Figure 6 and provides a very satisfying result, with an accuracy measured with the coefficient of determination in crossvalidation of $R^2_{FRIDA-linear} = 0.904$, significantly higher than the $R^2_{Ie} = 0.739$ scored by the State Index, and a set of weights of immediate physical interpretability reported in Table 2. By inspecting the weights, one can notice that those assigned to the predictors Flow and $SPEI_6$ are very low, although not null, and the index trajectory is mainly determined by Storage and Piezometer values. S1 and Pz2, in fact, describe the trajectories of the main water reservoirs of the region, lakes and groundwater, whose fluctuations are the result of natural variability as well as human regulation, mainly for irrigation purposes.

**Table 2.** Weights of the linear model calibrated on the optimal subset of predictors. The predictor Moy (month of the year), providing a seasonal information, is not directly included in the weights optimization but it is accounted for by depurating the variables of their annual cyclo-stationary mean.

| Predictor | Weight |
| --- | --- |
| Moy | / |
| Storage (S1) | 0.721 |
| Flow (F3) | $10^{-9}$ |
| Piezometer (P2) | 0.278 |
| $SPEI_6$ | $10^{-9}$ |

As a further analysis, we reiterated the model calibration and crossvalidation steps with a more complex, highly flexible model class, the ELM architecture, which scored an accuracy of $R^2_{FRIDA-ELM} = 0.907$. On the one hand, the arguably insignificant 0.005% improvement in accuracy of ELM with respect to the linear class, probably does not justify the loss of immediacy and transparency induced by the transition to a black-box model. On the other hand, this experiment proves the robustness of the linear model in constituting the model class of choice for this drought index. In table 3 we report a more detailed comparison between State index, FRIDA-linear and FRIDA-ELM indexes with several accuracy metrics. The analysis of other metrics seem to reinforce the conclusions drawn by considering $R^2$ only: both FRIDA indexes (linear and ELM) outperform the State Index quite significantly, while the difference among them is negligible, although the non-linear index is always the top performing.

The reported metrics do not distinguish between errors above and below the target deficit. Indeed, we consider these two error types of comparable importance. On the one hand, the underestimation of a deficit value may find the water users unprepared

**Table 3.** Accuracy of the State Index, FRIDA linear, and FRIDA ELM in reproducing the supply deficit, quantified in terms of coefficient of determination $R^2$, the Pearson correlation coefficient, the Root Mean Square Error (RMSE), and the fourth grade Root Mean Square Error (R4MS4E).

| Metric | State Index | Frida Linear | Frida ELM |
|--------|-------------|--------------|-----------|
| $R^2$ | 0.7396 | 0.9036 | 0.9074 |
| Pearson | 0.8601 | 0.9506 | 0.9533 |
| RMSE | 0.2066 | 0.1135 | 0.1014 |
| R4MS4E | 0.2549 | 0.1475 | 0.1299 |

5    to face a serious drought. On the other hand, the overestimation of drought conditions may ignite repeated false alarms that will compromise the index trustworthiness and its efficacy in triggering an alert state. Therefore, rather than penalizing an error above or below the target trajectory, we find more compelling to focus on errors in the most crucial drought situations i.e., at the maximum level of deficit recorded. One way of doing so is considering R4MS4E, as in Table 3, which penalizes errors in the deficit peaks. Another specific assessment tool for analyzing the indexes performance during critical droughts is the confusion

10    matrix, reporting the classification performance of critical droughts, here arbitrarily defined as months reporting deficit values above the 85th percentile (Tables 4, 5, 6). The rows of the confusion matrix represent the instances in a predicted class while the columns represent the instances in an actual class. Consequently, the main diagonal reports the number of correctly classified points. Cells outside the main diagonal specify the errors: the value in the bottom-left cell (first column, second row) indicates a situation in which the index does not recognize an ongoing drought, while the value in the top-right cell (first row and second column) indicates the number of false alarms. FRIDA-ELM confusion matrix seems to significantly exceed the competitors' performances by erroring only 0,57% of the times, as opposed to the 10,91% of $I_e$, and the 6,3% of FRIDA-linear.

**Table 4.** State Index confusion matrix.

| SI-deficit | critical drought | normality |
|------------|------------------|-----------|
| **critical drought** | 131 | 18 |
| **normality** | 1 | 24 |

**Table 5.** FRIDA-Linear confusion matrix.

| Frida Linear-deficit | critical drought | normality |
|----------------------|------------------|-----------|
| **critical drought** | 138 | 11 |
| **normality** | 0 | 25 |

**Table 6.** FRIDA-ELM confusion matrix.

| Frida Linear-deficit | critical drought | normality |
|---|---|---|
| critical drought | 147 | 2 |
| normality | 1 | 24 |

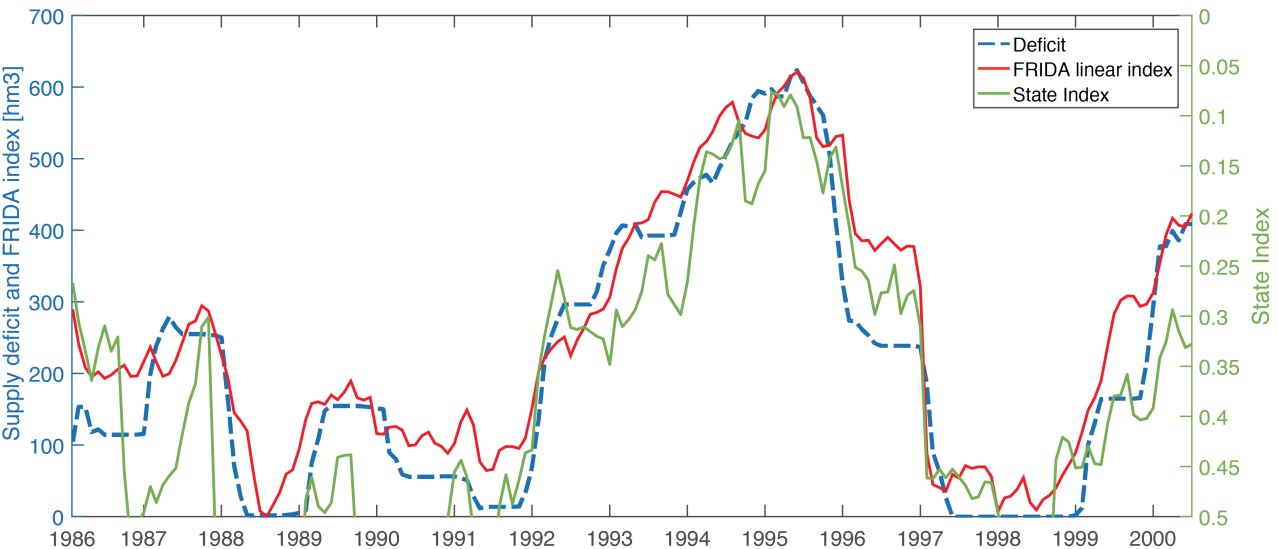

**Figure 6.** Comparison between the FRIDA linear index (blue) and the state index (green) in reproducing the monthly aggregated supply deficit (red). FRIDA index presents an higher similarity with the deficit and only requires 5 inputs instead of the 12 required by the state index.

## 5 Conclusions

The purpose of this study is to contribute to the identification of drought management strategies able to improve the efficiency
and resilience of drought prone regulated water systems. This problem is considered urgent as the analysis of climate trends shows that drought frequency and severity are intensifying all over in Europe, particularly in the Mediterranean area.

This work explores the potential of drought indexes as a management tool for the purpose of containing drought impacts. Since traditional indicators are often inadequate to characterize water availability conditions in highly regulated contexts, a novel framework for the construction of customized basin-specific drought indexes is proposed. This framework relies on the
10 employment of a feature extraction technique, the Wrapper for Quasi Equally Informative Subset Selection (W-QEISS). Given a set of information collected in the basin, W-QEISS features a deep learning machine that automatically selects the most suitable input set for the construction of a model reproducing the target variable, i.e., a ground truth representative for the state of water resources in the basin. Specifically, W-QEISS performs the search process in a four-dimensional metric space

of predictive accuracy, cardinality, relevance, and redundancy. On top of that, W-QEISS algorithm is designed to identify one subset with the highest predictive accuracy and multiple subsets with similar information content (i.e., quasi equally informative subsets). This provides insights on the relative relevance of the variables and a deeper understanding of the underlying physical processes taking place in the basin. The choice of the preferred input set and model class balance accuracy and practicality of the index. The efficacy of FRIDA methodology is strongly dependent on data availability, in terms of predictors diversity and numerosity, and length of the time series. FRIDA is best applicable in contexts where an extensive monitoring system has been in place for long enough to allow a consistent and informative dataset for the index calibration. However, while some hydro-meteorological variables are easy to monitor and most often available (e.g., precipitation, temperature), the accessibility of soil moisture, groundwater table level, snowpack extent, air humidity etc., may represent a problem. When a key drought-driving variable for the context at hand is absent from the input set, the efficacy of FRIDA is undermined.

The application of the FRIDA in the Jucar river basin case study has successfully demonstrated the suitability of the frame-work to design a basin specific drought index. Firstly, the automatic variable selection yields an immediate and informative result, which presents strong similarities with the empirical expert-based variable set employed by the CHJ, while involving a significantly lower number of features (5 variables instead of the 12 required by the State Index). Secondly, the newly computed FRIDA linear index outperforms the official Spanish State Index in terms of accuracy in reproducing the target variable, while maintaining immediate interpretability.

However, one of the reasons why the $I_e$ enjoyed such wide acceptance among the Jucar stakeholders is related to the widely comprehensive approach employed for its construction. All water users, in fact, feel represented in the index through at least one variable being observed in the proximity of their water related activity, even if such variable is low-weighted or redundant when computing the basin-wide aggregated indicator. The FRIDA approach does not ensure such representation of all water users, although it appears as a more rigorous and efficient alternative to the inclusive CHJ approach. Moreover, FRIDA is a portable methodology, suitable for the many drought prone contexts in need of a drought management plan. In conclusion, the aim of arranging an effective framework for the construction of basin customized combined drought indexes can be considered achieved. The indexes constructed with FRIDA have proven to be an asset for (i) representing drought conditions in highly regulated basins, where traditional indexes tend to fail; (ii) gaining a deeper understanding of the hydro-meteorological processes taking place in the basin; and (iii) constituting a valid alternative to the Spanish approach for the State Index design, thus supporting appropriate drought management strategies, such as triggering drought restraining response measures.

The already valid results achieved by this study open new possibilities for the use of basin-specific drought indexes. Fur-ther research efforts could be addressed to exploring the potential of employing FRIDA indexes in directly informing water management operations. Additionally, the possibility of forecasting such indexes can be tested in order to timely prepare for upcoming dry seasons. We expect that the projection of a drought index fosters the adoption of a proactive (as opposed to the current reactive) approach in facing a drought. Proactivity promotes a shift from costly and often belated mitigation measures, to preventive actions that will grant flexibility to timely prepare to upcoming droughts, while reducing costs associated to drought impacts and restrictions.

Ultimately, FRIDA can represent an asset for improving the system resilience under a changing climate. Despite the fact that FRIDA is conditioned upon historical data, one can imagine that in the short term, drivers' interactions and relative role in causing a drought hold unchanged. In this case, the index formulation remains valid in the context of a changing climate. In the long term, nevertheless, this hypothesis may cease to hold, we thus suggest a frequent reiteration of FRIDA procedure to monitor the evolution of drivers and dynamics leading to a drought in the basin. For example, in a groundwater dominated system as the Jucar basin, the piezometer information is likely to remain essential in a future climate, but, at the same time, we can expect evapotranspiration processes to increase their drought-propelling role, as climate change induces a general increase of temperatures. In other contexts, e.g., snow dominated catchments, the role of snow may lose priority due to a diminishing winter snowpack reserve. FRIDA will thus represent a valuable tool to support the analysis on the dynamic role of drivers in drought evolution under a changing climate.

*Code and data availability.* The complete dataset employed for the feature selection step can be downloaded open source from http://doi.org/10.5281/zenodo.1185084 (Zaniolo et al., 2018). A detailed description of FRIDA, including both data and codes, is available at http://www.nrm.deib.polimi.it/?page_id=2438.

*Acknowledgements.* The work has been partially funded by the European Commission under the IMPREX project belonging to Horizon 2020 framework programme (grant n. 641811). The authors would like to thank the Planning Office of the Confederación Hidrográfica del Júcar (CHJ) for providing the data used in this study.

## Abbreviations

**CHJ** Confederación Hidrográfica del Jucar.

**DMP** Drought Management Plan.

**ELM** Extreme Learning Machines.

**FRIDA** FRamework for Index-based Drought Analysis.

**Ie** Índice de Estado.

**IVS** Input Variable Selection.

**MOEA** Multi-Objective Evolutionary Algorithm.

**R4MS4E** Fourth grade Root Mean Square Error.

**RMSE** Root Mean Square Error.

**SPEI** Standardized Precipitation and Evapotranspiration Index.

**SPI** Standardized Precipitation Index.

**SRI** Standardized Runoff Index.

**W-QEISS** Wrapper for Quasi Equally Informative Subset Selection.

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
