# Peer review of "Automatic design of basin-specific drought indexes for highly regulated water systems"

_Hydrology and Earth System Sciences, 2017_

## Referee Comment (RC1) · Anonymous Referee #1 · 6 Dec 2017

The study presents a new framework for determining basin drought indicators (target index) by coupling numerous models conditioned to select and weight hydro-meteorological variable states (predictors) in an automated fashion. The manuscript is topically of interest and relevance to HESS readers, generally well written, and logically presented. Most comments and suggestions provided request clarification in the manuscript, although some additional (minor) analysis is perhaps warranted. Comments below.

1. Introduction: What is the motivation for selecting these 4 objectives, other than 'common' or 'convenient'? Additional justification or rationale is warranted.

2. Methods: Why is only f4 (accuracy) selected to discriminate among subsets (Fig 2, step 2)? Why this one and perhaps not others as well? Subsequently, all 4 objectives /

assessment metrics are used for presumably final selection. Does this ultimately indicate that accuracy is the most important objective? Or somehow give it more weight?

3. Results: The target variable (supply deficit) requires a more clear description earlier in the manuscript. A later statement (p17, L421) indicates that agricultural demand is used to computer the target deficient, however there could be many definitions (deficit in reservoir storage, deficit in long-term groundwater levels, deficit in meeting total demand, etc.) Why is ag demand used?

4. Results: A linear model is ultimately selected although a non-linear mode is recommended and compared. In terms of R2, there is little difference, however it may also be interesting to compare the weights given to each input/predictor. If there is a significant difference, this may not be intuitive (a statistical modeling artifact?)

5. Results: Other comparison metrics (between SI and FRIDA linear model) besides R2 may be warranted. How does the RMSE (or other) compare? Is it better to error above or below the target deficit?

6. Results: What are the FRIDA results using the exact set of 12 indicators included in the State Index? And associate weights? This may be useful for comparison (and discussion with water users.)

7. Conclusion: The authors make mention of a changing climate. What does this mean for the reliability and accuracy of the framework conditioned on historical (relatively stationary) data? Please discuss.

8. Discuss: If a subset of the 4 objectives are selected, or different objectives, how might this change the outcomes?

9. Discuss: What influence may the selection of the learning machine, MOEA algorithm, etc. have on outcomes? Are they sensitive to choices or not?

10. Discuss: Would the selected inputs/predictors change substantially is the target deficit were defined differently? It may not be overly surprising that reservoir volume

and groundwater levels are most important for a target deficit focused on agricultural irrigation demand.

11. What are the prospects for projecting out the State Index, based on the state of some features (e.g. reservoir volume) and predictions of other features (e.g. precipitation or recharge)? This is hinted at in the very end of the manuscript, but may warrant more discussion.

---

## Referee Comment (RC2) · Anonymous Referee #2 · 1 Feb 2018

The manuscripts provides an excellent contribution to the field for characterizing basin-specific drought conditions within a powerful framework that offers automation, replicability and flexibility. This is particularly useful in applying the approach in management (and planning) decisions at various temporal and spatial scales including reservoir operation, hydropower generation and water allocation among various users and the environment. An fine review on drought types, commonalities and differences is a good compendium to cite for research and educational purposes. The two algorithms presented in the selection of predictors, target variable and index subsets is a great contribution to the field which is often dominated by standardized indicators of droughts that may lack relevance in a local basin context where other confounding factors including regulations, water rights, environmental constraints and long-time operation rules merit

representation. That said, the manuscript would benefit from a better presentation of results, minor editorial improvements and some more detailed explanation of some of the calculations involved in estimating the index. In what follows, I provide some recommendations for improvement.

Major Issues

1. One of the major issues in the manuscript is presentation of the final step, the drought index. There is a strong disconnect between what is presented in figure 2 and the calculated index? It is clear the at the linear model was a a balanced way to obtain the supply deficit and 'the index'. Is the automated index the supply deficit? Fitness is really good compared to the well-established State Index but how it all fits together considering the different units and how are things calculated? This might seem like an unnecessary question but it is important to present with clarity the fundamental outcome of the approach .

2. Likewise for Table 2, how are these weights applicated? Elaborate on the exclusion of Moy in the weight and how is brought back so is taken into account. IF this is too much detail for the main paper consider an appendix for 1 and 2 above.

3. The set of conclusions are succinct and useful. However, I would highly recommend to comment before (or as part of these) the cases in which this approach may not be suitable. What are the challenges in obtaining predictors and developing computations, and where the approach presented in the paper which is actually promising moving in the field.

4. Perhaps offer an online supplementary material section in which users can play with the approach. I found it very suitable for an educational setting and in helping basins worldwide in organizing information to characterize drought. Even when some of the algorithms require a fair amount of training from the users, having a pre-processed repository would be of great service to the community.

5. From what I understand, supply deficit is the target variable. Description of it and its connection with the indicators of the basin is poor. So I encourage the authors to improve it in the paper to make it easier to follow how do we go from predictors, Pareto optimal sets, to index estimation (see 1 above) and and tests.

Minor Issues

1. I through revision of the abbreviations/acronyms in the paper is recommended. Examples: MOEA, ELM, CHJ.

2. Abstract, explain how it that traditional drought indexes fail to detect events. Not in the abstract but in the opening of the paper or the contributions section of the paper.

3. paragraph line 25, Why is the Jucar index superior to other approaches? What is the basis for comparison?

4. LIne 33, are the $100 billion for all Europe, over the time period? What does this mean in terms of GDP or other indicators? PUt some context to it otherwise is useless. What sectors are included what type of impacts?

5. line 37 what is meant by economic damage?

6. A graphic showing the four types of drought described would be very useful although not the main objective of the paper. Spatial, temporal, supply and demand, and involved sectors in a basin could be outlaid in the infographic.

7. Sentence starting in line 93 is awkward please break into more sentences.

8. The equation below line 230, should it be $f4(Si) >= f(Sj)$?

9. Line 320 as per comment above, elaborate on Ie performance.

10. How would a 'traditional index' e.g. SDI would perform in Figure 5? How are we making the case of both Ie and the developed automated index are better? Please elaborate.

---

## Author Comment (AC1) · 15 Feb 2018

The study presents a new framework for determining basin drought indicators (target index) by coupling numerous models conditioned to select and weight hydro-meteorological variable states (predictors) in an automated fashion. The manuscript is topically of interest and relevance to HESS readers, generally well written, and logically presented. Most comments and suggestions provided request clarification in the manuscript, although some additional (minor) analysis is perhaps warranted. Comments below.

1. Introduction: What is the motivation for selecting these 4 objectives, other than 'common' or 'convenient'? Additional justification or rationale is warranted.

[Figure]

FRIDA procedure bases its objectives formulation on information theory, as suggested in Taormina et al., (2016). We consider the use of such objectives combination the most suitable to design an operational index, while providing informative insights about the dynamics driving drought evolution in the basin. In particular, the maximization of accuracy ensures a precise reproduction of the data, while the minimization of cardinality aims at simplifying the final models. These characteristics are key for an operational index, expected to balance precision and ease-of-use.

Relevance and redundancy objectives are instead an asset for an effective subset search process as they foster the diversification of the solutions explored within the MOEA algorithm, while guaranteeing low intra-subset similarity and high information content of the solutions. In particular, a predictor can be strongly relevant, when its removal from the input set causes a significant drop in the model accuracy; irrelevant, when its presence or absence from the input set does not affect the model accuracy; and weakly relevant, when there exists a combination (namely, a Markov blanket) of other predictors carrying analogous information about the target variable (Yu and Liu, 2004). An optimal subset is thus composed of strongly relevant features, and non-redundant weakly relevant features. A weakly relevant predictor is non-redundant when its Markov blanket is not included in the input subset. Depending on the problem at hand, various combinations of weakly-relevant predictors can exist, producing quasi-equally informative models, and requiring the optimization of relevance and redundancy objectives to be entirely identified (Liu et al., 2015).

Following the reviewer suggestion, we will clarify the rationale behind the objectives selection in the Methods and tools chapter, specifically in section 2.2.

2. Methods: Why is only f4 (accuracy) selected to discriminate among subsets (Fig 2, step 2)? Why this one and perhaps not others as well? Subsequently, all 4 objectives /assessment metrics are used for presumably final selection. Does this ultimately indicate that accuracy is the most important objective? Or somehow give it more weight?

The objectives of relevance and redundancy are essential to support the search process towards the finding of a diversified and comprehensive set of solutions, which will not be achieved optimizing cardinality and accuracy only. A two-objective search based on cardinality and accuracy would, in fact, identify optimal solutions, but at the same time disregard a number of quasi-equally informative subsets with an almost identical operational behavior. The identification of such alternative solutions, nevertheless, grants flexibility and multiple options for the expert-based choice of the preferred subset, where certain combinations of predictors can be favored according to case-specific operative purposes, such as a more robust or less costly data gathering process, or enhanced acceptability and immediacy of the index.

Once the search is completed, although, the specific relevance and redundancy score of each variable combination is rarely of interest for the design of an operational index, while its accuracy and cardinality are crucial. When the dataset of candidate variables presents significant redundancy and correlation among features, numerous subsets characterized by a wide range of cardinalities are generally available to achieve a relative small range of accuracies. This is often the case in environmental problems, where spatial and temporal correlation of hydro-meteorological variables and associated indicators is significant. For this reason, the accuracy metric is initially used to discriminate among subsets, in order to limit the number of solutions that undergoes a deeper examination to the highly accurate solutions, provided they feature different cardinalities and predictors combinations.

We will discuss this point in the Methods section of the revised manuscript.

3. Results: The target variable (supply deficit) requires a more clear description earlier in the manuscript. A later statement (p17, L421) indicates that agricultural demand is used to computer the target deficient, however there could be many definitions (deficit in reservoir storage, deficit in long-term groundwater levels, deficit in meeting total demand, etc.) Why is ag demand used?

In a report issued by the Jucar Hydrological Confederation (CHJ, 2007) the Ie is validated against the supply deficit with respect to agricultural and urban demand, the validation result is considered satisfying, and the index is declared operative. The choice of employing the same supply deficit as a target variable for the FRIDA index is thus required to ensure comparability between the two indexes. As the reviewer suggests, we will add a sentence clarifying this point earlier in the manuscript.

4. Results: A linear model is ultimately selected although a non-linear mode is recommended and compared. In terms of R2, there is little difference, however it may also be interesting to compare the weights given to each input/predictor. If there is a significant difference, this may not be intuitive (a statistical modeling artifact?)

The black-box nature of the non-linear ELM model does not allow an analysis of the weight given to each predictor. ELM belongs to a family of models called Artificial Neural Networks (ANN), typically employed in regression problems and labeled universal approximators. In ANN models, the input features are manipulated through one or more layers of multiple hidden neurons performing sigmoidal transformations. The optimal number of layers, and of neurons in each layer, is problem dependent and generally needs manual calibration. The neurons' output is then weighted and summed to compose the final output. As a result of the non-linear manipulations, it is not possible to determine the weights assigned to each predictor in a ANN regression, and a comparison with the linear model is not possible in these terms.

5. Results: other comparison metrics (between SI and FRIDA linear model) besides R2 may be warranted. How does the RMSE (or other) compare? Is it better to error above or below the target deficit?

We consider the point of the reviewer well taken. In the revised version of the paper

we will report additional metrics to aid the comparison between indexes. In particular we will include the Pearson coefficient and the RMSE alongside R2. As it is evident from the result reported below, both FRIDA indexes (linear and ELM) outperform the State Index quite significantly, while the different among them is negligible, although the non-linear index is always the top performing.

We consider erroring above or below the target deficit of equal importance as, on the one hand, the underestimation of a deficit value may find the water users unprepared to face a serious drought. On the other hand, the overestimation of drought conditions may ignite repeated false alarms that will compromise the index trustworthiness and its efficacy in triggering an alert state. Therefore, rather than penalizing an error above or below the target trajectory, we find more compelling to focus on errors in the most crucial drought situations i.e., at the maximum level of deficit recorded. In order to do so, we included two additional metrics: R4MS4E, penalizing errors in the deficit peaks, and a confusion matrix reporting the classification performance of critical droughts, arbitrarily defined as months reporting deficit values above the 85th percentile. The rows of the confusion matrix represent the instances in a predicted class while the columns represent the instances in an actual class. Consequently, the main diagonal reports the number correctly classified points. Outside the diagonal the errors are reported: the value in the first column and second row indicates a situation in which the index does not recognize an ongoing drought, while the value in the first row and second column indicates the number of false alarms. These additional metrics substantially confirm the previously obtained results, with the exception of the FRIDA-ELM confusion matrix, that seems to significantly exceed the competitors' performances by erroring only 0.57% of the times, as opposed to the 10,91% of Ie, and the 6,3% of FRIDA-linear.

We will add the newly computed metrics in the paper in the form of tables (Tables: 1, 2, 3, 4 ), as reported below.

6. Results: What are the FRIDA results using the exact set of 12 indicators included in the State Index? And associate weights? This may be useful for comparison (and discussion with water users.)

[Figure]

**Table 1.** Accuracy metrics

| Metric | State Index | Frida Linear | Frida ELM |
|--------|-------------|--------------|-----------|
| Pearson | 0.8601 | 0.9506 | 0.9533 |
| R2 | 0.7396 | 0.9036 | 0.9074 |
| RMSE | 0.2066 | 0.1135 | 0.1014 |
| R4MS4E | 0.2549 | 0.1475 | 0.1299 |

**Table 2.** Confusion matrix State Index

| SI-deficit | critical drought | normality |
|------------|------------------|-----------|
| critical drought | 131 | 18 |
| normality | 1 | 24 |

**Table 3.** Confusion matrix Frida Linear

| Frida Linear-deficit | critical drought | normality |
|----------------------|------------------|-----------|
| critical drought | 138 | 11 |
| normality | 0 | 25 |

**Table 4.** Confusion matrix FRIDA-ELM

| Frida Linear-deficit | critical drought | normality |
|----------------------|------------------|-----------|
| critical drought | 147 | 2 |
| normality | 1 | 24 |

A linear model calibrated with the whole set of State Index inputs produces a very similar result to the FRIDA model in terms of accuracy, as it is evident from the metrics reported below (Table 5. This result is not surprising if we analyze the weights assigned by the calibration to each input: all negligible except for the storage and piezometer predictors included in the FRIDA selected subset (see Table 6). The use of the 12-predictors thus seems to bring no advantage, as on the one hand it complexifies the model, the data retrieval process, and the index computation by adding unnecessary predictors; and on the other hand it compromises the tool's acceptability. The official SI, in fact, is the outcome of a participatory process where variables and weights were negotiated with stakeholders. In particular, the weights carry a specific physical meaning as they are proportional to the demand class associated to each partial Ie (as detailed in section 3 of the paper). Thus, redefining the input weights will invalidate the outcome of the participatory process, while providing no benefit from an operational and modeling viewpoint.

**Table 5.** Performance of the Linear Model calibrated with the whole set of State Index inputs

| Metric | Linear Model |
| --- | --- |
| Pearson | 0.9505 |
| R2 | 0.9013 |
| RMSE | 0.1188 |
| R4MS4E | 0.1499 |

7. Conclusion: The authors make mention of a changing climate. What does this mean for the reliability and accuracy of the framework conditioned on historical (relatively stationary) data? Please discuss.

In the short term, one can imagine that, despite a change in the drivers' statistics due to climate change, their interactions and relative role in causing a drought holds un-

**Table 6.** Weights assigned to each predictor of the Linear Model calibrated with the whole set of State Index inputs

| Predictor | Weight |
|---|---|
| **S1** | **0.826** |
| S2 | 2.04E-10 |
| F1 | 2.04E-10 |
| F2 | 1.96E-10 |
| F3 | 3.55E-10 |
| F4 | 2.06E-10 |
| Pl1 | 3.17E-10 |
| Pl2 | 5.72E-10 |
| Pl3 | 2.82E-10 |
| Pz1 | 2.66E-10 |
| **Pz2** | **0.174** |
| Pz3 | 2.37E-09 |

changed. In this case, the index formulation remains valid in the context a changing climate. In the long term, nevertheless, this hypothesis may cease to hold, thus requiring the reiteration of FRIDA procedure to identify new drivers and dynamics leading to a drought in the basin.

For example, in a groundwater dominated system as the Jucar basin, the piezometer information is likely to remain essential in a future climate, but, at the same time, we can expect evapotranspiration processes to increase their drought-propelling role, as climate change induces a general increase of temperatures. In other contexts, e.g., snow dominated catchments, the role of snow may lose priority due to a diminishing winter snowpack reserve.

FRIDA will thus represent a valuable tool to support a thorough analysis on the role of each driver in drought evolution under a changing climate.

We consider this a good point and we will expand the paper's conclusion to clarify the matter.

8. Discuss: If a subset of the 4 objectives are selected, or different objectives, how might this change the outcomes?

FRIDA procedure bases its objectives formulation on information theory, as suggested in Taormina et al., (2016). Information theoretic criteria (e.g., SU, Mutual information, and Partial Mutual Information) do not assume any functional relationship between the variables and thus result well suited to quantify the dependence between two variables in any modeling context (McKay, 2003). Other objectives formulations could be explored, for instance substituting the use of Symmetric Uncertainty with more traditional correlation coefficients, although with the risk of losing generality by assuming linear dependence between variables. The use of a subset of objectives could, in principle, be a viable option in case of a two-objectives search using accuracy and cardinality only. Such optimization will require less computational time, but on the other hand, will return a poorer set of solutions with respect to the four-objective search (see answer at point 3 for more details). We will include this comment in the Methods and tools section of the revised version of the paper.

9. Discuss: What influence may the selection of the learning machine, MOEA algorithm, etc. have on outcomes? Are they sensitive to choices or not?

The Extreme Learning Machine and Borg MOEA were selected as they were proven to perform well under a suite of different problems, ensuring applicability, scalability, accuracy and, in the case of ELM, very limited computational burden. ELM bypasses the time consuming gradient-based search of optimal neurons parameters required by traditional ANN techniques, by performing a random selection of hidden nodes, followed by the optimization of their output weights. Such optimization is solved through a one-step matrices product and essentially amounts to learning a linear model. However, we do not expect the choice of the learning machine or MOEA to be crucial for the attainment of the result. A different benchmark MOEA (e.g., NGSAII, MOEAD,

eps-MOEA) is likely to come to a comparable result, although requiring a possibly significant effort in the manual calibration of the evolution parameters, which is automated in Borg MOEA. Similarly, other ANN techniques could in principle be substituted to ELM, although running the risk of incrementing the computational time to unbearable levels, given the multiple calibration and validation processes reiterated in WQEISS. We will include this comment in the Methods and Tools section of the revised paper.

10. Discuss: Would the selected inputs/predictors change substantially is the target deficit were defined differently? It may not be overly surprising that reservoir volume and groundwater levels are most important for a target deficit focused on agricultural irrigation demand.

The selection of the target variable is indeed a critical step for the FRIDA procedure and requires an expert consultation to select the most appropriate target for the basin, and the operational aim of the index. We agree with the reviewer that the result of the variable selection step is not surprising given the basin climate and the physical meaning of the target variable. However, the aim of the paper was to demonstrate the validity of a completely automated procedure (i.e., that requires no information on system topology or basin characteristics) in recognizing the main drought drivers, and predicting a deficit with accuracy and limited computational effort. The selection of the case study was tailored to this purpose, as the Jucar basin successfully relies on a drought index to activate restraining measures. The analysis of the Jucar State Index provided guidelines for our work, firstly in terms of target choice and candidate variable retrieval, and secondly for validating FRIDA in both the variable selection step and index design outcome.
We thank the reviewer for bringing up this point, and we will specify the matter in the revised version of the paper.

11. What are the prospects for projecting out the State Index, based on the state of some features (e.g. reservoir volume) and predictions of other features (e.g.

precipitation or recharge)? This is hinted at in the very end of the manuscript, but may warrant more discussion.

Following the reviewer suggestion, we will expand the closing sentence of the paper to provide some clarification on the matter. We expect that the projection of a drought index fosters the adoption of proactive (as opposed to the current reactive) approach in facing a drought. Proactivity translates in a shift from costly and often belated mitigation measures, to preventive actions, thus granting flexibility to timely prepare to upcoming droughts, while reducing costs associated to drought impacts and restrictions.

---

## Author Comment (AC2) · 15 Feb 2018

The manuscripts provides an excellent contribution to the field for characterizing basin-specific drought conditions within a powerful framework that offers automation, replicability and flexibility. This is particularly useful in applying the approach in management (and planning) decisions at various temporal and spatial scales including reservoir operation, hydropower generation and water allocation among various users and the environment. An fine review on drought types, commonalities and differences is a good compendium to cite for research and educational purposes. The two algorithms presented in the selection of predictors, target variable and index subsets is a great contribution to the field which is often dominated by standardized indicators of droughts that may lack relevance in a local basin context where other confounding factors including

regulations, water rights, environmental constraints and long-time operation rules merit representation. That said, the manuscript would benefit from a better presentation of results, minor editorial improvements and some more detailed explanation of some of the calculations involved in estimating the index. In what follows, I provide some recommendations for improvement.
We thank the referee for the positive comment

Major Issues

1. One of the major issues in the manuscript is presentation of the final step, the drought index. There is a strong disconnect between what is presented in figure 2 and the calculated index? It is clear the at the linear model was a a balanced way to obtain the supply deficit and 'the index'. Is the automated index the supply deficit? Fitness is really good compared to the well-established State Index but how it all fits together considering the different units and how are things calculated? This might seem like an unnecessary question but it is important to present with clarity the fundamental outcome of the approach.
The supply deficit is identified as target variable to guide the construction of the automated index, the index is therefore a proxy of the deficit, not the deficit itself. The supply deficit, in fact, was not obtained by linear model, but was simulated with the AQUATOOL model (Andreu et al., 1996), a Decision Support System developed at the Universidad Politécnica de Valencia (UPV), Valencia, Spain. The model can run in simulation mode with a monthly time step, and it is conceived in the form of a flow network with different types of oriented connections that reproduce water losses, hydraulic connections between nodes, reservoirs and aquifers, and flow limitations based on elevation. Within AQUATOOL, complex processes such as evaporation and infiltration are effectively reproduced. The modeled supply deficit represents the monthly nominal shortage of water conveyed to the irrigation districts, and is only quantifiable a posteriori, when the water shortage has already jeopardized the fields. On the other hand, the automated index can be constantly monitored, and thus represents a valuable management tool for containing drought impacts and identifying effective drought management strategies. Numerical results show that FRIDA methodology outperforms the benchmark State Index in the representation of the recorded deficit, and, to assess so, we employed the coefficient of determination R2. R2 is calculated as the ratio of the explained variance (the proportion to which a mathematical model reproduces the dispersion of a given data set) to the total variance, and is a common measure of correlation. As a consequence, the unit in which a variable is expressed has no impact in the computation of R2 and the values of correlation reported are perfectly comparable. We will remark the above points in the text.

2. Likewise for Table 2, how are these weights applicated? Elaborate on the exclusion of Moy in the weight and how is brought back so is taken into account. IF this is too much detail for the main paper consider an appendix for 1 and 2 above.
In the linear case, the index is calculated as a weighted sum of the form:
$$Index = weight1 * predictor1 + weight2 * predictor2 + weight3 * predictor3 + (\ldots)$$
The predictor Moy represents the succession of the months in the year, and is an expression of the seasonality of hydro-meteorological processes. Not surprisingly, it is selected as a relevant variable in the feature selection step. Moy is constructed as the repetition of an array of numbers from 1 to 12 for the length of the considered time horizon, and thus presents a discontinuous shape: a slow and steady increase followed by a steep decrease in correspondence to the onset of a new year. While the non-linear models employed in the feature selection can effortlessly handle such an intermittent vector, linear models struggle with similar shapes. We therefore decided to account for the seasonality in the linear model indirectly, i.e., excluding Moy from the predictors set, but consistently considering seasonality by depurating the predictors of their annual cyclostationary mean.
Following the reviewer suggestion, we will clarify the matter in the text.

3. The set of conclusions are succinct and useful. However, I would highly recommend to comment before (or as part of these) the cases in which this approach may not be

suitable. What are the challenges in obtaining predictors and developing computations, and where the approach presented in the paper which is actually promising moving in the field.

We thank the referee for the good point raised. We find this comment in accordance to the suggestions of referee 1, and, accordingly, we will expand the conclusion section of the paper to substantially improve its clarity, and better elaborate on the mentioned issues.

The efficacy of FRIDA methodology is strongly dependent on the data availability, in terms of predictors diversity and numerosity, and length of the time series. FRIDA is best applicable in contexts where an extensive monitoring system has been in place for long enough to allow a consistent and informative dataset for the index calibration.  However, while some hydro-meteorological variables are easy to monitor and most often available (e.g., precipitation, temperature), the accessibility of other variables such as soil moisture, groundwater table level, snowpack extent, air humidity etc., may represent a problem. When a key drought-driving variable for the context at hand is absent from the input set, the efficacy of FRIDA is undermined. Concerning the computation, FRIDA procedure is mostly automatized. The only step that may require a small degree of manual calibration is the number of ELM to be employed in the feature selection step. An inadequate number of ELM may yield a poor result due to insufficient learning potential (too few ELM), or tendency to overfit the data (too many ELM).

4. Perhaps offer an online supplementary material section in which users can play with the approach. I found it very suitable for an educational setting and in helping basins worldwide in organizing information to characterize drought. Even when some of the algorithms require a fair amount of training from the users, having a pre-processed repository would be of great service to the community.

Following the referee suggestion, we will set up an online repository including all the candidate predictors and the AQUATOOL modeled deficit. A readme text file will detail the nature of the inputs, their sources, and units of measures, as well as the experimental settings adopted in our experiments. For the codes needed to run the experiments we will refer to the publicly available W-QEISS repository.

5. From what I understand, supply deficit is the target variable. Description of it and its connection with the indicators of the basin is poor. So I encourage the authors to improve it in the paper to make it easier to follow how do we go from predictors, Pareto optimal sets, to index estimation (see 1 above) and and tests.

The supply deficit is the monthly shortage of water conveyed to the irrigation districts with respect to their nominal water demand. As such, it represents the manifestation of a drought as perceived by the water users, i.e., an operational drought. As detailed in the introduction, an operational drought is driven by two factors: water availability due to hydro-meteorological fluctuations, and management of available water resources. An operational drought index must, on the one hand, account for hydro-meteorological predictors, and on the other hand, detect situations of water shortage as perceived by users, i.e., supply deficit. We will make sure this message is conveyed in the introduction of the paper, in order to clarify the relation between supply deficit and operational indicators. In step 1 of FRIDA, possibly relevant hydro-meteorological predictors are identified, and in step 2 those predictors are combined to define Pareto optimal sets. Providing multiple optimal sets we allow the customization of the index, foreseeing its employment as a management tool for the constant monitoring of water resources. The decision maker can identify the preferred subset in accordance to context-dependent needs of accuracy, and agility in the variables monitoring. Finally, in step 3 of the Framework, we tested the potential in constructing meaningful operation drought indexes. To do so, we opted for one of the Pareto optimal sets, we calibrated the relative index on the target deficit, and tested its performance in terms of accuracy and cardinality against the benchmark state index. We here remark that the state index was validated on the same supply deficit, granting a fair comparison between the two indexes. Following the reviewer suggestion, we will make sure to clarify the workflow leading to index design in the section dedicated to present the FRIDA framework.

Minor Issues

1. I through revision of the abbreviations/acronyms in the paper is recommended. Examples: MOEA, ELM, CHJ.
We will extend the table of acronyms to include every acronym included in the manuscript.

2. Abstract, explain how it that traditional drought indexes fail to detect events. Not in the abstract but in the opening of the paper or the contributions section of the paper.
Meteorological, agricultural, and hydrological indexes fail in representing drought conditions in highly regulated basins, where the presence of man-managed water infrastructures (lake dams, groundwater pumps) filter water availability and have a role in magnifying or restraining drought impacts. On the other hand, traditional operational drought indexes are often designed to operate analysis over coarse spatiotemporal resolutions, thus resulting unsuitable for a real time basin level drought detection, characterization, and management. Highly regulated systems need ad hoc index formulations to account for uncontrolled hydro-meteorological conditions (precipitation, temperature...) as well as controllable variables (reservoir and groundwater levels).

3. paragraph line 25, Why is the Jucar index superior to other approaches? What is the basis for comparison?
The State index is a well-established drought monitoring tool currently in use in every Spanish hydrographic confederation (Jucar, Duero, Segura, Ebro, Guadalquivir, Tajo...) to monitor the state of water resources in the basin and trigger drought restraining measures when certain threshold values of the index are reached. Each confederation has designed its customized formulation for the state index which reflects the hydroclimatic conditions and the water uses of the region, and is the outcome of a long participatory process involving basin experts and stakeholders. Since their establishment in 2007, the State indexes have represented the most consistent and extensively applied example of index used for drought management purposes. Thus, Ie represent the state of the art for basin-customized operational drought indexes, and a remarkable

benchmark to test the proposed FRIDA methodology.
We will elaborate on the choice of the State Index as a benchmark in the paper introduction.

4. Line 33, are the $100 billion for all Europe, over the time period? What does this mean in terms of GDP or other indicators? Put some context to it otherwise is useless. What sectors are included what type of impacts?
As stated in the EU "drought and water scarcity second interim report" (European Commission, 2007), economic impacts of drought amount to $100 billion for the period 1976-2006. This figure was estimated by aggregating information provided by EU member states on the economic impacts of drought, which, as the report states in page 32, include impacts endured by consumers and households, tourism, industry, energy, and agriculture. No information is provided in terms of GDP units.

5. line 37 what is meant by economic damage?
By economic damage caused by drought we refer to a situation in which a water deficit induced by droughts affects production, sales and business in a variety of sectors (Spinoni et al., 2016). The main economic impacts divided by sectors are detailed in the same report are: socioeconomic impacts; impacts on environmental, forestry, wildfires, and biodiversity; impacts on farming and livestock; impacts on public water supply; impacts on surface and groundwater; impacts on industry; impacts on power generation: hydropower, thermal, and nuclear; impacts on commercial shipping; impacts on tourism and recreation. In the introduction of the manuscript we will mention the matter and refer to Spinoni et al., 2016 for further details.

6. A graphic showing the four types of drought described would be very useful although not the main objective of the paper. Spatial, temporal, supply and demand, and involved sectors in a basin could be outlaid in the infographic.

We will include the infographic in Figure 1 in the paper introduction.

Caption: Development chain of droughts through time. Meteorological drought: defined as a lack of precipitation over a region for a certain period of time; develops in the short term. Agricultural drought: accounts for the plants and crops water stress; develops in the medium term. Hydrological drought: defined as a period of low streamflow in watercourses, lakes and groundwater level below normal; develops in the long term. Operational drought: defined as a period with anomalous supply failures in a developed water exploitation system. Figure adapted from Spinoni et al., 2016 to include Operational drought.

7. Sentence starting in line 93 is awkward please break into more sentences.
We will rephrase as:
Anthropized systems have, in fact, demonstrated the ability to endure meteorological droughts for months or even years without suffering consequences i.e., without incurring in a situation of water shortage perceived by users. The employment of an effective planning and management of water resources system enables such systems to wisely exploit the combined storage capacities of surface and groundwater reserves and restrain drought.

8. The equation below line 230, should it be $f4(Si) >= f(Sj)$?
There is indeed an error in the formula, we thank the reviewer for noticing. The correct formula is
$\mathbf{S}_i \subset \mathbf{S}_j$ and $f_4(\mathbf{S}_i) \geq f_4(\mathbf{S}_j)$.
. We will make sure to fix it in the revised version of the paper.

9. Line 320 as per comment above, elaborate on Ie performance.
According to the reviewer suggestion we will remark the significance of the Spanish experience with State Indexes in this instance, in accordance to what elaborated in the response to minor issue 3.

10. How would a 'traditional index' e.g. SDI would perform in Figure 5? How are we making the case of both Ie and the developed automated index are better? Please

elaborate.

We hereby assume the reviewer intended to mention the traditional SPI index.  SPI would represent meteorological conditions but won't provide any insight on actual water shortage as perceived by the users. Both Ie and Frida index are operational indexes, and take into account the filtering effect of water management on meteorological fluctuations, thus correlating significantly with the recorded supply deficit.

REFERENCES:

CHJ, 2007. Anejo2 - Plan especial de alerta y eventual sequia en la confederacion hidrografica del Jucar. Confederacion Hidrografica del Jucar, Jucar River Basin Management Authority, Ministry of Agriculture, Food and Environment, Spanish Government, Valencia, Spain (in Spanish). European Commission, 2007. Water Scarcity and Droughts, Second Interim Report, June 2007, http://ec.europa.eu/environment/water/quantity/pdf/comm_droughts/2nd_int_report.pdf. Accessed 09 February 2018.

Andreu, Joaquin, Jy Capilla, and Emilio Sanchis. AQUATOOL, a generalized decision-support system for water-resources planning and operational management. Journal of hydrology 177.3-4 (1996): 269-291.

Spinoni, J., Naumann, G., Vogt, J., Barbosa, P. (2016). Meteorological Droughts in Europe: Events and Impacts – Past Trends and Future Projections. Publications Office of the European Union, Luxembourg, EUR 27748 EN, doi:10.2788/450449.

Taormina, R., Galelli, S., Karakaya, G., Ahipasaoglu, S., 2016. An information theoretic approach to select alternate subsets of predictors for data-driven hydrological models. Journal of Hydrology 542, 18–34.

**Fig. 1.** Drought Evolution infographic